# On the Necessity of Step-Specific Representation Learning for Multi-Step Forecasting

## Abstract

This paper demonstrates that modern deep forecast models are susceptible to a fundamental EXPRESSIVENESS BOTTLENECK, which stems from the use of step-invariant representations in multi-step prediction tasks. Through theoretical analysis, we demonstrate that step-invariant representation causes an unavoidable forecast error that cannot be overcome simply by advancing neural architectures. To address this issue, we propose Step-wise Representation adaPtation (SRP): first, a foundation model is pre-trained for one-step-ahead forecast; subsequently, the model is adapted to various forecast horizons via low-rank adaptation. This design enables the generation of step-specific representations, thereby avoiding the expressiveness bottleneck. Moving forward, we further establish SRP++, which employs adaptively weighted low-rank adapters to mitigate the expressiveness bottleneck while enhancing efficiency and forecast performance. Experiments show that SRP++ significantly improves model expressiveness and outperforms state-of-the-art time-series forecast methods. Code is available at https://anonymous.4open.science/r/SRP-7C55.

## 1 Introduction

Time-series forecast seeks to predict future values of a sequence from its historical observations and constitutes a core component of real-world applications, such as finance (*e.g.*, stock selection (Hu et al., 2025)), meteorology (*e.g.*, weather forecast (Allen et al., 2025)), and manufacturing (*e.g.*, process monitoring (Wang et al., 2023a)). Recent advances in deep learning have led to the widespread adoption of deep forecast models. Central to these approaches is a fundamental question: *How to extract discriminative representations from historical sequences truly helpful for forecast.*

Most contemporary research tackles this question by devising specialized neural architectures that model temporal dynamics to obtain representations. Representative examples span recurrent neural networks (Salinas et al., 2020), convolutional neural networks (Wu et al., 2023; Wang et al., 2023b), and graph neural networks (Yi et al., 2023a). Current progress is characterized by an ongoing debate between Transformers and simple linear models. Transformers, powered by self-attention mechanisms, provide superior scalability (Liu et al., 2024; Nie et al., 2023; Piao et al., 2024), whereas linear models, which capture temporal dynamics using linear layers, are straightforward to implement and often perform competitively (Zeng et al., 2023; Yang et al., 2024a; Lin et al., 2024). These advancements showcase the rapid evolution of representation learning in time-series analysis.

Despite above architectural progress, existing methods predominantly rely on a step-invariant (SI) representation. They typically use neural networks to encode historical sequence into a single representation vector, then use a linear layer to generate forecasts for all future steps. **In this work, we reveal that this step-invariant representation constitutes a fundamental expressiveness bottleneck.** Heuristically, it falsely assumes that the optimal representations for predicting all future steps are identical. Operationally, it forces a shared representation across all future steps, which constrains the predictions to simple linear combinations of this shared representation. Theoretically, we prove that this constraint imposes an unavoidable forecast error that cannot be overcome by advancing neural architectures.

A natural solution to the expressiveness bottleneck is to learn step-specific representations for multi-step forecasting. To this end, we propose Step-wise Representation adaPtation (SRP), a principled two-stage framework designed to generate step-specific representations. SRP begins by pre-training

a model for one-step-ahead forecasting, followed by adaptation to longer forecasting horizons using the low-rank adaptation (LoRA) technique Hu et al. (2021). However, a straightforward implementation of this framework incurs significant computational costs and may yield suboptimal performance, as it fails to model the dependencies among different forecast steps. To overcome these limitations, we introduce SRP++, which leverages a mixture-of-experts mechanism to enable selective parameter sharing across step-specific LoRA modules. This design not only reduces computational overhead but also captures the relationships between forecast steps, effectively improving both efficiency and predictive accuracy.

Our main contributions are summarized as follows:

- We identify and theoretically prove the expressiveness bottleneck hampering and being inherent in modern deep forecast models.
- We propose the SRP framework to address the bottleneck via step-specific representations. We further develop SRP++, which is augmented by adaptively weighted low-rank adapters exploit dependencies between forecast steps to enhance both efficiency and accuracy.
- We conduct comprehensive experiments and demonstrate that SRP++ can effectively improve diverse forecast models over different public datasets.

## 2 PRELIMINARIES

In this work, we consider the multi-step time-series forecast problem: predicting future observations from historical data. As a preface, it is important to emphasize a key distinction between two prevailing paradigms in multi-step forecast: iterative forecast, where future values are predicted sequentially by feeding previous forecasts back as inputs, and direct forecast, where the entire sequence of future values is predicted in a single pass Wang et al. (2025). **Our focus in this study is on the direct forecast paradigm**, which has currently become dominant in deep forecast models.

Formally, consider a time-series dataset $X$ with D covariates, where $X_n$ denotes the observation at the $n$-th step. We define two central constructs (Box et al., 2015): (1) **historical sequence** $L = [X_{n-L+1}, \ldots, X_n] \in \mathbb{R}^{L \times D}$, where L is the historical window length; (2) **label sequence** $Y = [X_{n+1}, \ldots, X_{n+T}] \in \mathbb{R}^{T \times D}$, where T is the forecast horizon. Based on these elements, the task can be described as estimating $\mathbb{E}[Y|L]$, the expected label sequence conditioned on the history (Nguyen et al., 2024; Ghimire et al., 2024).

Most deep forecast models comprise two primary components. Firstly, a neural encoder $g_e$ extracts informative representations from historical sequence, producing $R = g_e(L)$. Secondly, a linear header $g_d$ transforms this representation to generate the forecast: $\hat{Y} = g_d(R)$. The learnable parameters in $g_e$ and $g_d$ are optimized to minimize the discrepancy between $\hat{Y}$ and $Y$. Previous work has largely focused on advancing encoder architectures, leading to the development of diverse frameworks such as convolutional neural networks (Wu et al., 2023; Wang et al., 2023b), graph neural networks (Yi et al., 2023a), and Transformers (Liu et al., 2024; Vaswani et al., 2017).

## 3 PROPOSED METHOD

### 3.1 MOTIVATION

Modern deep forecast models are prevalently trained to generate the full sequence simultaneously (Liu et al., 2024; Wu et al., 2023; Zeng et al., 2023). In this section, we demonstrate that this approach suffers from an expressiveness bottleneck due to its reliance on step-invariant representations, causing an unavoidable forecast error that cannot be overcome simply by advancing neural architectures.

Let $R \in \mathbb{R}^{L \times D}$ be the encoder output. The direct forecast approach produces forecasts using a linear layer with learnable weights $W \in \mathbb{R}^{T \times L}$ and $b \in \mathbb{R}^T$:

$$\left[\hat{Y}_1, \ldots, \hat{Y}_T\right] = [W_1, \ldots, W_T] R + [b_1, \ldots, b_T]. \tag{1}$$

where $\hat{Y}_t = W_t R + b_t$ is the prediction for the $t$-th future step. This assumes a single step-invariant representation $R$ is optimal for all forecast steps, and that different linear transformations suffice to

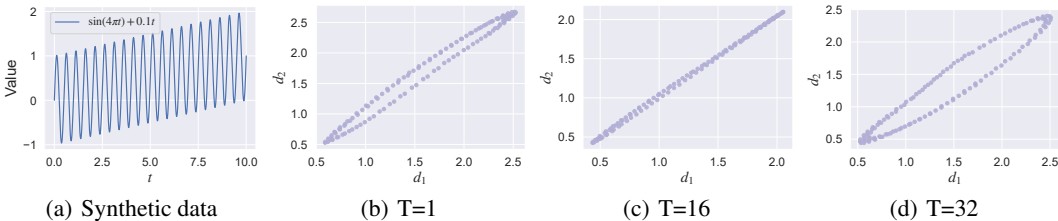

(a) Synthetic data  (b) T=1  (c) T=16  (d) T=32

Figure 1: Visualization of step-specific representations. (a) Synthetic time series; (b–d) 2D encoder features (pre-decoder) for forecast steps T=1, 16, 32. Each step trains a separate single-output MLP with history length L=16.

generate accurate predictions across all horizons. However, this assumption becomes problematic when different forecast steps require fundamentally different representational features, particularly in long-term forecast scenarios where T is large. The reliance on step-invariant representations creates an *expressiveness bottleneck*: regardless of encoder quality, using a shared representation across all forecast steps leads to an unavoidable modeling error. As shown in Theorem 3.1, this error is strictly positive when $L + 1 < T$ — a typical setting in long-term forecast tasks.

**Theorem 3.1** (Expressiveness Bottleneck). *Let $\bar{W} = [W\ b] \in \mathbb{R}^{T \times (L+1)}$ be the parameters in the SI's linear decoder, $Y \in \mathbb{R}^{T \times D}$ be the label sequence; the minimum attainable estimation error is*

$$\|\epsilon\|_F^2 = \sum_{t=\mathrm{rank}(\bar{W})+1}^{T} \|U_t^\top Y\|_2^2, \tag{2}$$

*where $\bar{W} = U\Sigma V^\top$ is the singular value decomposition of $\bar{W}$, $\mathrm{rank}(\bar{W}) \leq \min\{T, L+1\}$, and $\{U_i\}_{i=\mathrm{rank}+1}^{T}$ form an orthonormal basis for the null space of $\bar{W}$. Notably, this error is independent of the representation $R$ provided by encoder.*

**Case Study.** To illustrate how step-invariant representations limit expressiveness, we conduct a case study on synthetic data. We set the historical window length to $L = 16$ and use a two-layer perceptron encoder (hidden sizes 16 and 2) with a single-output linear decoder. For each forecast step, we train a separate model and visualize the resulting representations before the decoder layer in Figure 1 (b)-(d). The results show that optimal representations differ significantly across steps, demonstrating that enforcing step-invariant representations creates an expressiveness bottleneck that leads to unavoidable modeling errors.

Given the substantial limitations inherent in step-invariant representations, there is a clear need for forecast strategies that can harness the power of step-specific representations. This raises two key questions: *How to design a framework that generates and exploits step-specific representations for forecast? Does step-specific representation truly improve forecast performance?*

### 3.2 STEP-WISE REPRESENTATION ADAPTATION

According to Theorem 3.1, the error disappears when the linear layer's output dimension is 1. Motivated by this, we propose Step-wise Representation adaPtation (SRP), a two-stage framework that transitions from step-invariant to step-specific representation to address the expressiveness bottleneck.

**Pre-training.** We begin by training a foundation model for one-step prediction. Given the historical sequence $L \in \mathbb{R}^{L \times D}$, the process can be described as follows:

$$R = g_e(L), \hat{Y}_1 = g_d(R) = WR + b, \tag{3}$$

$$\mathcal{L}_{\mathrm{pre}} = \|Y_1 - \hat{Y}_1\|_2^2, \tag{4}$$

where the encoder $g_e$ extracts features $R$, and the one-step-ahead prediction $\hat{Y}_1 \in \mathbb{R}^D$ is generated using a single-output linear projection with weights $W \in \mathbb{R}^{1 \times L}$ and bias $b \in \mathbb{R}$. The pre-training

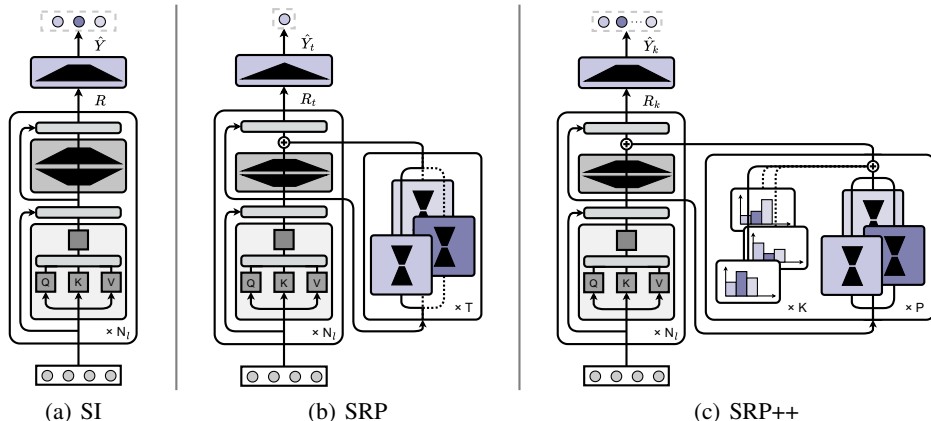

(a) SI      (b) SRP      (c) SRP++

Figure 2: Visualization of SI, SRP, and SRP++ approaches. Gray blocks denote identical encoder components. Purple blocks represent decoding strategies. Rectangles with varying transparencies indicate different expert matrices in SRP++.

objective is the MSE on the one-step prediction, establishing a solid foundation for subsequent step-specific adaptations.

**Adaptation.** After pre-training for one-step-ahead prediction ($T = 1$), we adapt the model for multi-step forecast ($T \geq 2$). For each forecast step $t$, we introduce step-specific LoRA modules to selected encoder linear layers, enabling the generation of step-specific representations $R^{(t)}$ while freezing the weights of the original encoder and decoder. Specifically, for a selected linear layer with weight matrix $M \in \mathbb{R}^{d_{out} \times d_{in}}$, we modify the weights for each step $t$ as follows:

$$M^{(t)} = M + B^{(t)}A^{(t)}, \tag{5}$$

where $B^{(t)} \in \mathbb{R}^{d_{out} \times r}$ and $A^{(t)} \in \mathbb{R}^{r \times d_{in}}$ are LoRA matrices, with rank $r < \min(d_{out}, d_{in})$. For each forecast step $t$, the modified encoder $g_e^{(t)}$ generates step-specific predictions as follows:

$$R^{(t)} = g_e^{(t)}(L), \hat{Y}_t = g_d(R^{(t)}) = WR + b, \tag{6}$$

$$\mathcal{L}_{ada}^{(t)} = \|Y_t - \hat{Y}_t\|_2^2, \tag{7}$$

where the adaptation objective is the MSE on the $t$-step prediction. Notably, only the LoRA matrices $(B^{(t)}, A^{(t)})$ are optimized to minimize the adaptation objective, while the rest of the foundation model remains frozen. This process is repeated for each prediction step $1 < t \leq T$, enabling the model to generate forecasts for diverse steps while avoiding the expressiveness bottleneck.

**Discussion.** SRP overcomes the expressiveness bottleneck by transitioning from step-invariant to step-specific representations. By ensuring both the pre-trained and adapted models use single outputs, SRP eliminates the error term identified in Theorem 3.1. The step-specific representations $R^{(t)}$ are specifically optimized for each forecast step $t$, removing the constraint that forces all predictions to be linear combinations of the same representation bases. Moreover, during the adaptation stage, the pre-trained weights remain frozen, preventing any interference with the foundation model. This property has a critical implication for time-series forecast: adapting to long-range forecasts does not compromise the performance on short-term predictions.

### 3.3 SRP++: Contextualizing SRP for Forecasting

Time series data exhibits strong autocorrelation, inducing correlations across multiple forecast steps. However, the SRP approach treats each forecast step independently by assigning separate LoRA matrices to generate step-specific representations and using a single-output layer for prediction. This leads to two limitations: (1) the number of adaptations increases linearly with the forecast horizon,

resulting in high computational cost; and (2) neglecting inter-step correlations can impair forecast accuracy despite achieving step-specific representations.

In this section, we propose SRP++, which leverages the dependencies among forecast steps for improved performance and efficiency. According to Theorem 3.1, the expressiveness bottleneck vanishes as long as the number of outputs does not exceed $L + 1$; thus, *single-step output is not strictly required*. Building on this insight, we partition the T-step label sequence into K contiguous segments, each containing S = T/K steps. The foundation model's output dimension is then increased from 1 to S, allowing each adaptation to generate step-specific representations for S future steps jointly, and thereby reducing the number of required adaptations. This segment-based adaptation exploits the step-wise dependency within each segment to improve accuracy and efficiency, while maintaining step-specific expressiveness as long as $S \leq L + 1$.

To further exploit dependency across segments, we enable partial parameter sharing across different LoRA modules via a mixture-of-experts mechanism. Specifically, suppose we have P expert matrices, each denoted as $B^{(p)}$ and $A^{(p)}$. For a selected linear layer in the encoder with original weights $M$, the step-specific adaptation for segment $k$ is given by

$$M^{(k)} = M + \sum_{p=1}^{P} \Delta_k^{(p)} \cdot B^{(p)} A^{(p)}, \tag{8}$$

where $\Delta_k = [\Delta_k^{(1)}, \Delta_k^{(2)}, \ldots, \Delta_k^{(P)}]$ is a learnable, normalized weight vector that adaptively combines the P expert matrices for each segment. This design enables expert matrices to be shared across segments, allowing the model to leverage inter-segment dependencies in the label sequence and enrich supervision signals for improved step-specific adaptation. A visual comparison among step-invariant (SI), SRP, and SRP++ approaches is provided in Figure 2.

## 3.4 OVERALL WORKFLOW

In this section, we detail the procedure for applying SRP++ to train step-specific forecast models, as outlined in Algorithm 1. The workflow demonstrates how to transition from step-invariant to step-specific representations while leveraging mixture-of-experts principles for efficiency. The workflow begins by pre-training a foundation model for S-step prediction (step 1). After pre-training, the foundation model parameters are frozen (step 2). Next, we inject step-specific adaptation modules into the backbone encoder: for each of the K segments, a segment-specific weight vector $\Delta_k$ and P shared LoRA expert modules $\{B^{(p)}, A^{(p)}\}_{p=1}^{P}$ are introduced. For each segment $k$, the mixing weights $\Delta_k$ are learned to adaptively combine the shared LoRA experts, as defined in (8) (step 5). During adaptation, the segment-specific weight vector and segment-shared LoRA modules are optimized using the corresponding segment's ground truth, while the backbone remains frozen (steps 6-8). During inference, the segment-specific adapters are applied to the

---

**Algorithm 1** The workflow of SRP++.

**Input**: $X$: training dataset, K: number of segments, P: number of LoRA experts, $r$: rank of LoRA matrices.
**Parameter**: $\{\Delta_k\}_{k=1}^{K}$: segment-specific weight vectors, $\{B^{(p)}, A^{(p)}\}_{p=1}^{P}$: LoRA expert matrices.
**Output**: $\{\Delta_k\}_{k=1}^{K}, \{B^{(p)}, A^{(p)}\}_{p=1}^{P}$: optimized adaptation parameters.

1: Pre-train a S-step model $\{g_e, g_d\}$ on $X$
2: Freeze the foundation model parameters
3: **for** $k = 1$ to K **do**
4:     **for** $L, Y$ in $X$ **do**
5:         Modify the encoder $g_e$ to $g_e^{(k)}$:
        $M^{(k)} \leftarrow M + \sum_{p=1}^{P} \Delta_k^{(p)} \cdot B^{(p)} A^{(p)}$
6:         $R^{(k)} \leftarrow g_e^{(k)}(L), \hat{Y}_k \leftarrow g_d(R^{(k)})$
7:         $\mathcal{L}_{ada}^{(k)} \leftarrow ||Y_k - \hat{Y}_k||_2^2$
8:         Update $\Delta_k$ and $\{B^{(p)}, A^{(p)}\}_{p=1}^{P}$ with $\mathcal{L}_{ada}^{(k)}$
9:     **end for**
10:     Freeze the segment-specific weight vector $\Delta_k$
11: **end for**
12: Return optimized step-specific adaptation parameters

---

frozen foundation model to generate step-specific representations for each segment, which are then concatenated to produce the final estimations.

## 4 EXPERIMENTS

To validate the effectiveness of our SRP++ approach for time-series forecast, we conduct a comprehensive empirical evaluation across four key dimensions:

Table 1: Multi-step forecast performance.

| Models | SRP++ (Ours) | | iTransformer (2024) | | FreTS (2023) | | TimesNet (2023) | | TiDE (2023) | | DLinear (2023) | | FEDformer (2022) | | Autoformer (2021) | | Informer (2021) | | Transformer (2017) | | TCN (2017) | |
|---|---|---|---|---|---|---|---|---|---|---|---|---|---|---|---|---|---|---|---|---|---|---|
| Metrics | MSE | MAE | MSE | MAE | MSE | MAE | MSE | MAE | MSE | MAE | MSE | MAE | MSE | MAE | MSE | MAE | MSE | MAE | MSE | MAE | MSE | MAE |
| ETTm1 | **0.400** | **0.406** | 0.415 | 0.416 | 0.407 | 0.415 | 0.413 | 0.418 | 0.419 | 0.419 | 0.404 | 0.407 | 0.440 | 0.451 | 0.596 | 0.517 | 0.887 | 0.693 | 0.943 | 0.733 | 0.891 | 0.632 |
| ETTm2 | **0.287** | **0.330** | 0.294 | 0.335 | 0.335 | 0.379 | 0.297 | 0.332 | 0.358 | 0.404 | 0.344 | 0.396 | 0.302 | 0.348 | 0.326 | 0.366 | 1.256 | 0.801 | 1.322 | 0.814 | 3.411 | 1.432 |
| ETTh1 | 0.443 | **0.441** | 0.449 | 0.447 | 0.488 | 0.474 | 0.478 | 0.466 | 0.628 | 0.574 | 0.462 | 0.458 | **0.441** | 0.457 | 0.476 | 0.477 | 1.064 | 0.806 | 0.993 | 0.788 | 0.763 | 0.636 |
| ETTh2 | **0.377** | **0.401** | 0.390 | 0.410 | 0.550 | 0.515 | 0.413 | 0.426 | 0.611 | 0.550 | 0.558 | 0.516 | 0.430 | 0.447 | 0.478 | 0.483 | 4.358 | 1.719 | 3.296 | 1.419 | 3.325 | 1.445 |
| ECL | **0.174** | **0.265** | 0.176 | 0.267 | 0.209 | 0.297 | 0.214 | 0.307 | 0.251 | 0.344 | 0.225 | 0.319 | 0.229 | 0.339 | 0.228 | 0.339 | 0.335 | 0.416 | 0.274 | 0.367 | 0.617 | 0.598 |
| Traffic | **0.425** | **0.284** | 0.428 | 0.286 | 0.552 | 0.348 | 0.535 | 0.309 | 0.760 | 0.473 | 0.673 | 0.419 | 0.611 | 0.379 | 0.637 | 0.399 | 0.727 | 0.404 | 0.680 | 0.376 | 1.001 | 0.652 |
| Weather | 0.266 | **0.286** | 0.281 | 0.302 | **0.255** | 0.299 | 0.262 | 0.288 | 0.271 | 0.320 | 0.265 | 0.317 | 0.311 | 0.361 | 0.349 | 0.391 | 0.595 | 0.541 | 0.632 | 0.552 | 0.584 | 0.572 |
| PEMS03 | **0.112** | **0.222** | 0.116 | 0.226 | 0.146 | 0.257 | 0.118 | 0.223 | 0.316 | 0.370 | 0.233 | 0.344 | 0.174 | 0.302 | 0.501 | 0.513 | 0.137 | 0.241 | 0.126 | 0.233 | 0.666 | 0.634 |
| PEMS08 | **0.138** | **0.236** | 0.159 | 0.258 | 0.174 | 0.277 | 0.154 | 0.245 | 0.319 | 0.378 | 0.294 | 0.377 | 0.232 | 0.322 | 0.630 | 0.572 | 0.319 | 0.314 | 0.249 | 0.266 | 0.713 | 0.629 |

*Note*: We fix the input length as 96 following the established benchmarks (Liu et al., 2024; Wu et al., 2023). **Bold** typeface highlights the top performance for each metric, while underlined text denotes the second-best results. The results are averaged over prediction lengths (96, 192, 336 and 720), with full results in Appendix Table 2.

1. **Performance:** *How does SRP++ perform compared to current state-of-the-art step-invariant methods?* In Section 4.2, we benchmark SRP++ against state-of-the-art baselines on public datasets.

2. **Generality:** *Is it effective to enhance other forecast models?* In Section 4.3, we assess the versatility of applying SRP++ to enhance different forecast models.

3. **Flexibility:** *Does it support different adaptation methods?* Section 4.3 explores SRP++'s flexibility by substituting LoRA with alternative adaptation modules while maintaining step-specific capabilities.

4. **Sensitivity:** *How sensitive is SRP++ to hyperparameter changes?* In Section 4.4, we conduct a sensitivity analysis on the key hyperparameters.

## 4.1 EXPERIMENTAL SETUP

**Datasets.** We use nine standard time-series forecast datasets: ETT (with 4 subsets), ECL, Traffic, Weather, and PEMS (with 2 subsets), following the settings in (Wu et al., 2021) and (Liu et al., 2024). Each dataset is split chronologically into training, validation, and test sets. Appendix 4.1 provides detailed descriptions and statistics for each dataset.

**Baselines.** We select a diverse suite of baseline models for comparison: Transformer (Vaswani et al., 2017), Informer (Li et al., 2021), Autoformer (Wu et al., 2021), FEDformer (Zhou et al., 2022b), iTransformer (Liu et al., 2024), DLinear (Zeng et al., 2023), FreTS (Yi et al., 2023b), Times-Net (Wu et al., 2023) and TCN (Bai et al., 2018).

**Implementation details.** To benchmark models under a unified and fair protocol, we reproduced baselines using the official scripts from the TSLib[1]. All models were trained using the Adam optimizer (Kingma & Ba, 2015), tuning learning rates from $\{0.0001, 0.0005, 0.001\}$. The forecast horizons were set to $\{96, 192, 336, 720\}$ for ETT, ECL, Traffic, and Weather datasets, and to $\{12, 24, 36, 48\}$ for PEMS datasets. Performance was evaluated using mean squared error (MSE) and mean absolute error (MAE).

To apply SRP++ to enhance the baseline models, we first pre-trained baselines using their original hyperparameters with forecast horizons set to $T/K$ where $K \in \{2, 3, 4, 6\}$. During the subsequent step-specific adaptation stage, we tuned the learning rate ($\eta$) and SRP++ specific parameters (the LoRA rank $r$ and the number of LoRA experts). We conducted all experiments on NVIDIA RTX 4090 GPUs, with more implementation details provided in Appendix 4.2.

---

[1] https://github.com/thuml/Time-Series-Library

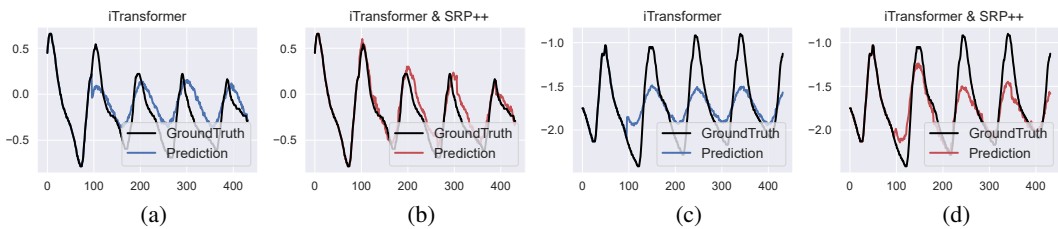

Figure 3: Visualization of forecast sequence generated with and without SRP++ under two snapshots.

Table 2: Varying adaptation technique results.

| Variants | SI | | | | Ada | | | | IA$^3$ | | | | SRP | | | | SRP++ | | | |
|---|---|---|---|---|---|---|---|---|---|---|---|---|---|---|---|---|---|---|---|---|
| Metrics | MSE | Δ | MAE | Δ | MSE | Δ | MAE | Δ | MSE | Δ | MAE | Δ | MSE | Δ | MAE | Δ | MSE | Δ | MAE | Δ |
| ETTh2 96 | 0.301 | - | 0.349 | - | **0.292** | 2.93%↓ | 0.342 | 2.10%↓ | 0.304 | 0.88%↑ | 0.346 | 0.84%↓ | 0.293 | 2.53%↓ | 0.342 | 2.13%↓ | 0.293 | 2.57%↓ | **0.341** | 2.25%↓ |
| ETTh2 192 | 0.382 | - | 0.402 | - | 0.384 | 0.55%↑ | 0.398 | 1.08%↓ | 0.381 | 0.17%↓ | 0.397 | 1.34%↓ | 0.380 | 0.54%↓ | 0.398 | 1.00%↓ | 0.377 | 1.25%↓ | **0.397** | 1.32%↓ |
| ETTh2 336 | 0.430 | - | 0.434 | - | 0.420 | 2.31%↓ | 0.431 | 0.78%↓ | 0.418 | 2.88%↓ | 0.428 | 1.48%↓ | 0.422 | 1.81%↓ | 0.433 | 0.23%↓ | 0.421 | 2.06%↓ | 0.429 | 1.22%↓ |
| ETTh2 720 | 0.447 | - | 0.455 | - | 0.430 | 3.87%↓ | 0.446 | 1.97%↓ | 0.437 | 2.23%↓ | 0.447 | 1.68%↓ | 0.428 | 4.25%↓ | 0.444 | 2.44%↓ | **0.418** | 6.49%↓ | **0.438** | 3.67%↓ |
| ETTh2 Avg | 0.390 | - | 0.410 | - | 0.382 | 2.05%↓ | 0.402 | 1.95%↓ | 0.385 | 1.28%↓ | 0.405 | 1.22%↓ | 0.381 | 2.31%↓ | 0.402 | 1.98%↓ | **0.377** | 3.33%↓ | **0.401** | 2.20%↓ |
| Weather 96 | 0.201 | - | 0.247 | - | 0.202 | 0.43%↑ | 0.246 | 0.39%↓ | 0.204 | 1.69%↑ | 0.249 | 0.80%↑ | 0.202 | 0.40%↑ | 0.245 | 1.00%↓ | **0.173** | 13.78%↓ | **0.211** | 14.71%↓ |
| Weather 192 | 0.250 | - | 0.283 | - | 0.248 | 0.83%↓ | 0.279 | 1.23%↓ | 0.250 | 0.14%↓ | 0.281 | 0.63%↓ | 0.248 | 0.90%↓ | **0.279** | 1.45%↓ | **0.246** | 1.43%↓ | 0.280 | 1.02%↓ |
| Weather 336 | 0.302 | - | 0.317 | - | 0.299 | 0.89%↓ | 0.317 | 0.03%↓ | 0.300 | 0.60%↓ | 0.315 | 0.66%↓ | 0.280 | 7.28%↓ | 0.297 | 6.44%↓ | **0.277** | 8.28%↓ | **0.296** | 6.63%↓ |
| Weather 720 | 0.370 | - | 0.362 | - | 0.361 | 2.41%↓ | 0.351 | 3.04%↓ | 0.365 | 1.40%↓ | 0.356 | 1.66%↓ | **0.357** | 3.57%↓ | **0.349** | 3.63%↓ | 0.367 | 0.86%↓ | 0.356 | 1.60%↓ |
| Weather Avg | 0.281 | - | 0.302 | - | 0.278 | 1.07%↓ | 0.299 | 0.99%↓ | 0.280 | 0.36%↓ | 0.300 | 0.66%↓ | 0.272 | 3.20%↓ | 0.293 | 2.98%↓ | **0.266** | 5.34%↓ | **0.286** | 5.30%↓ |

*Note*: Δ denotes the relative error improvement compared to iTransformer with SI paradigm.

## 4.2 OVERALL PERFORMANCE

The performance of our proposed step-wise representation adaptation approach on the MSTF task is presented in Table 1. We use iTransformer as the base model $g_\theta$ and apply it to different forecast horizons using the two-stage SRP++ framework. Overall, SRP++ significantly enhances the performance of iTransformer. For example, on the ETTm1 dataset, SRP++ reduces the MSE of iTransformer by 0.015. Similar improvements are observed across other datasets, underscoring the effectiveness of step-specific adaptation in overcoming the limitations imposed by shared representations across all forecast steps.

Importantly, SRP++ not only improves iTransformer's performance but also enables it to surpass models that previously outperformed iTransformer on certain datasets and metrics, like PEMS08 with MSE and MAE. This suggests that the gains achieved by SRP++ go beyond architectural designs alone, emphasizing the critical importance of addressing the expressiveness bottleneck through step-specific representations.

**Showcases.** To further illustrate the improvements provided by SRP++, we visualize forecast sequences for two snapshots from the ETTm2 dataset with a forecast horizon of T = 336 in Figure 3. While step-invariant approaches follow the general trends of the true label sequence, they struggle with capturing sharp peaks due to the expressiveness bottleneck, leading to misaligned forecasts across different horizons. In contrast, SRP++ mitigates this issue by employing step-specific representations tailored to each forecast horizon, resulting in more accurate predictions that track both trends and sharp peaks across multiple time steps.

## 4.3 GENERALIZATION STUDIES

In this section, we explore the generality of the SRP++ framework to enhance varying forecast models and encompass existing adaptation techniques.

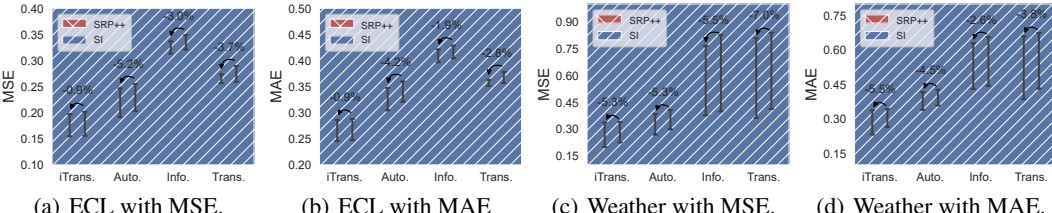

(a) ECL with MSE.  (b) ECL with MAE  (c) Weather with MSE.  (d) Weather with MAE.

Figure 4: Benefit of incorporating SRP++ in varying models, shown with colored bars for means over forecast lengths (96, 192, 336, 720) and error bars for 95% confidence intervals.

Table 3: Varying segment number performance.

| | ETTh1 | | | | Weather | | | |
|---|---|---|---|---|---|---|---|---|
| K | MSE | Δ | MAE | Δ | MSE | Δ | MAE | Δ |
| SI | 0.390 | - | 0.410 | - | 0.201 | - | 0.247 | - |
| 1 | 0.390 | 0.09% ↓ | 0.410 | 0.21% ↓ | 0.201 | 0.02% ↓ | 0.246 | 0.69% ↓ |
| 2 | 0.383 | 2.02% ↓ | 0.403 | 1.79% ↓ | 0.202 | 0.25% ↑ | 0.246 | 0.61% ↓ |
| 3 | 0.389 | 0.33% ↓ | 0.409 | 0.24% ↓ | 0.200 | 0.54% ↓ | 0.244 | 1.23% ↓ |
| 4 | **0.379** | 2.93% ↓ | **0.400** | 2.47% ↓ | 0.201 | 0.03% ↓ | 0.245 | 1.07% ↓ |
| 8 | 0.391 | 0.14% ↑ | 0.408 | 0.70% ↓ | **0.173** | 13.99% ↓ | **0.211** | 14.68% ↓ |
| 16 | 0.390 | 0.10% ↓ | 0.407 | 0.79% ↓ | 0.176 | 12.57% ↓ | 0.214 | 13.53% ↓ |
| 32 | 0.388 | 0.66% ↓ | 0.403 | 1.81% ↓ | 0.174 | 13.43% ↓ | 0.213 | 14.02% ↓ |
| 48 | 0.389 | 0.27% ↓ | 0.405 | 1.37% ↓ | 0.198 | 1.50% ↓ | 0.244 | 1.47% ↓ |
| 96 | 0.389 | 0.31% ↓ | 0.404 | 1.61% ↓ | 0.181 | 9.92% ↓ | 0.221 | 10.84% ↓ |

*Note*: Δ denotes the relative error improvement compared to iTransformer with SI paradigm. The results are generated with forecast length fixed at 96.

**Generalization to Adaptation Techniques.** We implement SRP++ by replacing the LoRA modules with two alternative adaptation techniques: Adapter (Houlsby et al., 2019a) (Ada) and IA$^3$ (Liu et al., 2022a) (IA$^3$), to demonstrate its support for existing adaptation techniques while preserving step-specific adaptation capabilities. Both Adapter and IA$^3$ are well-established parameter-efficient adaptation technologies. We also introduce a variant that applies step-specific adaptation with standard LoRA modules (SRP) for comparison. Detailed illustrations of their technical differences and parameter settings are provided in Appendix 5.3. The results in Table 2 indicate that all variants exhibit comparable improvements over step-invariant approaches, affirming SRP++'s flexibility in integrating diverse adaptation strategies while maintaining step-specific expressiveness. The standard SRP++, enabling parameter sharing across different segments with the mixture-of-experts mechanism, outperforms the variants since it leverages inter-step information while maintaining computational efficiency and step-specific capabilities.

**Generalization to Forecasting Models.** We incorporate SRP++ into several well-established forecast models: iTransformer, Autoformer, Informer, and Transformer. The results, averaged across different prediction lengths (96, 192, 336, 720) and accompanied by 95% confidence intervals, are presented in Figure 4. Overall, SRP++ improves the performance of these models by transitioning them from step-invariant to step-specific paradigms. Notably, Autoformer and Informer benefit significantly from SRP++, showing a relative MSE improvement of over 4% on both the ECL and Weather datasets. These results demonstrate the generality and broad applicability of SRP++, reinforcing its potential as a plug-and-play strategy for enhancing various neural forecast models.

## 4.4 SENSITIVITY STUDIES

In this section, we examine the impact of key hyperparameters on SRP's performance, with results shown in Table 3 and Figure 5. The main observations are as follows:

- The coefficient K determines the number of segments. We observe that SRP++ outperforms step-invariant approaches across nearly all values of K. The best performance is typically achieve at small K values (*e.g.*, K = 4 for ETTh1), demonstrating that finer-grained step-specific adaptations

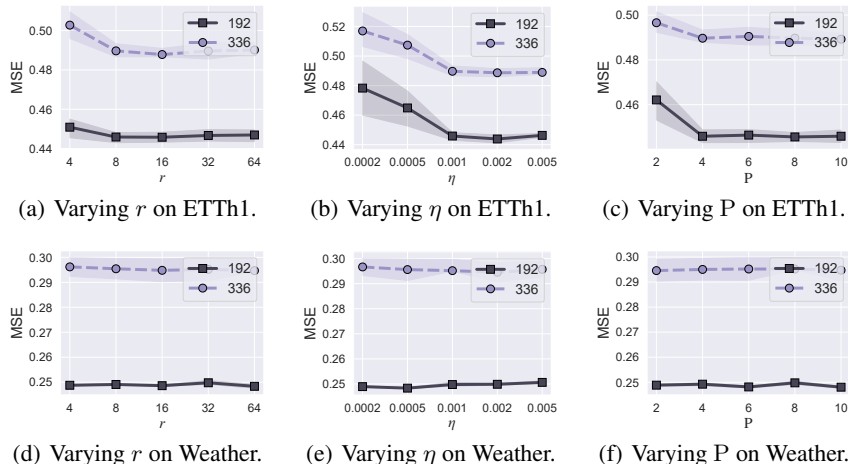

(a) Varying $r$ on ETTh1.  (b) Varying $\eta$ on ETTh1.  (c) Varying P on ETTh1.

(d) Varying $r$ on Weather.  (e) Varying $\eta$ on Weather.  (f) Varying P on Weather.

Figure 5: Performance given varying rank $r$, learning rate $\eta$ and the number of experts P.

are more effective in capturing step-specific patterns while still leveraging correlations in neighboring steps. Therefore, although ideally each prediction step would have its own step-specific representation (*i.e.*, K = T), the segmentation strategy offers a flexible solution that reduces computational complexity while maintaining the benefits of step-specific expressiveness.

- The coefficient $r$ determines the rank of the LoRA expert modules. We observe that both excessively small and large ranks degrade performance, with the optimal value balancing step-specific expressiveness and overfitting risks. For instance, on ETTh1, $r = 8$ achieves minimal MSE values of 0.441 (T = 192) and 0.484 (T = 336), while higher ranks overfit and lower ranks lack sufficient expressiveness for step-specific adaptation.

- The coefficient $\eta$ determines the learning rate in the adaptation phase. The model is sensitive to it, increasing $\eta$ from 0.0002 to 0.001 on ETTh1 reduces MSE from 0.459 to 0.440 for 192-step forecasts, which underscores the role of update rate in the adaptation phase.

- The coefficient P determines the number of LoRA expert modules. It exhibits stable performance unless it severely mismatches the segment count K. For example, on Weather (192-step), expanding experts from 2 to 10 only reduces MSE by 0.001, suggesting robustness to moderate variations while maintaining step-specific capabilities. These trends highlight the necessity of calibrating $r$ and $\eta$ precisely for effective step-specific adaptation, while the expert count offers flexibility within practical bounds.

## 5 CONCLUSION

In summary, this work identifies a fundamental expressiveness bottleneck in modern deep forecast models arising from step-invariant representations. Through rigorous theoretical analysis, we demonstrate that this limitation causes an unavoidable forecast error irrespective of architectural advances. To overcome this challenge, we propose SRP, a two-stage framework that enables step-specific representation learning via low-rank adaptation, and further extend this approach with SRP++, which introduces adaptively weighted adapters for improving efficiency and accuracy. Extensive experiments confirm that SRP++ substantially alleviates the expressiveness bottleneck and improves the performance of diverse forecast models over different public datasets.

**Limitations and Future Work.** In this study the parameter cost of LoRA still scales with model size and forecast horizon. Exploring alternative parameter-efficient adaptation techniques, such as prompt-tuning or adapter-tuning, may further enhance scalability and applicability. Additionally, the segmentation of forecast steps for SRP++ is manually specified. Automatically determining optimal segment boundaries, possibly via meta-learning techniques, represents a promising direction for further optimizing step-specific adaptation.

## REPRODUCIBILITY STATEMENT

The anonymous downloadable source code is available at `https://anonymous.4open.science/r/SRP-7C55`. For theoretical results, a complete proof of the claims is included in the Appendix C; For datasets used in the experiments, a complete description of the dataset statistics and processing workflow is provided in the Appendix D.1.

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

## A   RELATED WORK

### A.1   TIME SERIES FORECASTING MODELING

Time-series forecasting (TSF) modeling generally involves encoding historical sequences to extract discriminative representations for future predictions. To exploit temporal dynamics during encoding, various deep learning backbones have been developed, generally grouped into four main categories: RNN-based (*e.g.*, SegRNN (Lin et al., 2023)), CNN-based (*e.g.*, TimesNet (Wu et al., 2023)), GNN-based (*e.g.*, MAGNN (Chen et al., 2023)), MLP-based, and Transformer-based methods. Recent debates focus on MLP versus Transformer architectures, where MLPs (*e.g.*, DLinear (Zeng et al., 2023), TSMixer (Ekambaram et al., 2023)) are efficient but limited in handling complex temporal patterns, whereas Transformers (*e.g.*, PatchTST (Nie et al., 2023), iTransformer (Liu et al., 2024)) excel in temporal encoding but are computationally intensive. To better capture intricate temporal patterns, specialized designs such as series decomposition (*e.g.*, Autoformer (Wu et al., 2021)) and multiperiodicity analysis (*e.g.*, FiLM (Zhou et al., 2022a)) have been proposed, addressing seasonal and mixed period forecasting, respectively. Recent innovations explore frequency-domain representations for temporal patterns, exemplified by FedFormer (Zhou et al., 2022b) which employs frequency-domain attention score computation to enhance both efficiency and effectiveness. This paradigm demonstrates remarkable adaptability across architectures, including Transformers (Zhou et al., 2022b; Wu et al., 2021), MLPs (Yi et al., 2023b), and GNNs (Yi et al., 2023a; Cao et al., 2020), establishing frequency-domain analysis as a versatile plug-and-play component for temporal modeling.

Despite significant advancements in representation learning, modern TSF models predominantly rely on step-invariant (SI) representations, encoding historical sequences into a single shared representation for all forecast steps. This approach inherently suffers from an `expressiveness bottleneck`, where predictions are confined to linear transformations of a shared representation, limiting the model's capacity to capture step-specific temporal patterns. Our theoretical analysis reveals that this constraint induces unavoidable forecast errors, highlighting fundamental limitations in conventional TSF paradigms.

### A.2   MODULARIZED ADAPTATION

Adaptation has emerged as a pivotal technique for leveraging pre-trained models in downstream tasks, initially gaining prominence in natural language processing and computer vision (Xin et al., 2024). Modern approaches have evolved into modular frameworks categorized into three paradigms: Adapter-based, Selection-based, and LoRA-based methods. The Adapter-based paradigm introduces lightweight task-specific modules between pre-trained layers, preserving original parameters while enabling domain adaptation (Houlsby et al., 2019b; He et al., 2021; Lei et al., 2023; He et al., 2023b). In contrast, Selection-based methods employ parameter masking strategies to identify critical subnetworks for task-specific tuning (Liao et al., 2023; He et al., 2023a; Vucetic et al., 2022; Gheini et al., 2021). The LoRA-based paradigm marks a significant technical evolution in parameter-efficient adaptation by introducing Low-Rank Adaptation (LoRA) (Hu et al., 2021; Fomenko et al., 2024), which augments certain layers of a pre-trained model with trainable low-rank matrices. Instead of updating the full set of model parameters, LoRA only optimizes a small, low-rank decomposition of the weight update, thereby drastically reducing memory and computational overhead. This design allows for efficient adaptation to new tasks with minimal storage and enables fast switching between tasks by maintaining separate LoRA weights. LoRA's effectiveness and versatility have been further demonstrated through updated matrix decomposition (Yang et al., 2024b; Zhang et al., 2023), quantization (Dettmers et al., 2023; Li et al., 2023), and ranking adaptation (Ding et al., 2023; Xia et al., 2024), reflecting a broader trend toward modularized and scalable LoRA applications.

In the context of TSF, LoRA-based adaptation has demonstrated promising capabilities from two main perspectives. On one hand, (Nie et al., 2024) introduces channel-aware LoRA, leveraging low-rank adaptation to capture channel dependencies. On the other hand, a series of studies (Khanal et al., 2024; Chang et al., 2023; Zhou et al., 2023; Gupta et al., 2024) investigate the impact of LoRA within time series foundation models, focusing on how LoRA facilitates efficient adaptation and task transfer in time series forecasting. Though these explorations are promising, they largely overlook two critical aspects: the expressiveness bottleneck we have identified in TSF, and the untapped potential of LoRA for overcoming this limitation. Our work diverges by specifically addressing the

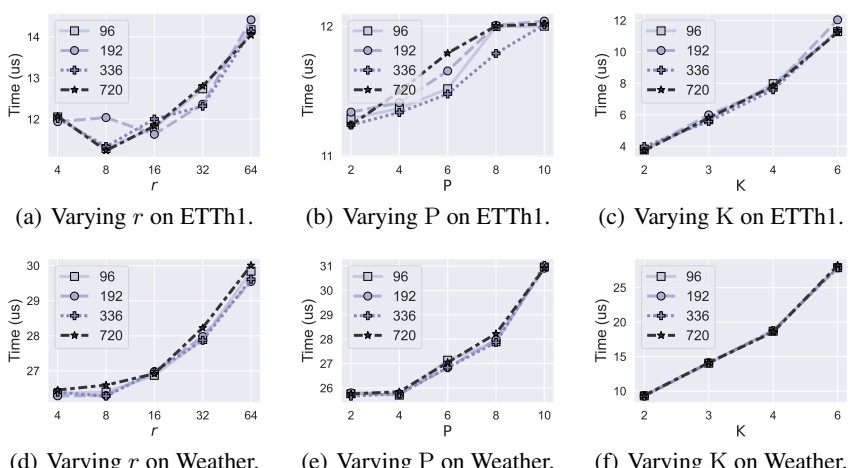

(a) Varying $r$ on ETTh1.  (b) Varying P on ETTh1.  (c) Varying K on ETTh1.

(d) Varying $r$ on Weather.  (e) Varying P on Weather.  (f) Varying K on Weather.

Figure 6: Running time of SRP++ given varying rank $r$, the number of experts P and the number of segments K.

expressiveness bottleneck through innovative integration of temporal segmentation and Mixture-of-LoRA enhanced adaptation, ensuring both model efficiency and step-specific expressiveness.

## B COMPLEXITY ANALYSIS

In this section, we conduct a parameter count analysis and evaluate the running cost of SRP++ through empirical investigation. Take iTransformer (Liu et al., 2024) as the base model, let the number of layers in iTransformer be $N_l$, the hidden dimension of the attention layer be $d_m$, and the hidden dimension of the FFN layer be $d_{ff}$. Then, the number of parameters introduced by SRP++ is given by:

$$
\begin{aligned}
N_{SRP++} &= N_l \times 2 \times (N_{LoRA} \times P + N_{Weight} \times K) \\
&= N_l \times 2 \times ((d_m \times r + r \times d_{ff}) \times P + P \times K),
\end{aligned}
\tag{9}
$$

where $N_{LoRA}$ and $N_{Weight}$ are the number of parameters introduced by LoRA expert module and learnable weights for each expert, respectively.

Consider that iTransformer is a standard Transformer architecture with dimension permutation in the input sequence, its number of parameters can be given by:

$$
\begin{aligned}
N_{iTrans} &= N_l \times (N_{Att} + N_{FFN} + N_{LN}) \\
&= N_l \times (4 \times d_m^2 + 2 \times d_m \times d_{ff} + 4 \times d_m),
\end{aligned}
\tag{10}
$$

where $N_{Att}$, $N_{FFN}$ and $N_{LN}$ are the number of parameters introduced by self attention layer, feedforward layer and layer normalization layer, respectively.

The ratio between the two parameter counts is therefore:

$$
\frac{N_{SRP++}}{N_{iTrans}} = \frac{P \times (d_m \times r + r \times d_{ff} + K)}{2 \times d_m \times (d_m + d_{ff} + 1)}.
\tag{11}
$$

For example, with $P = 4$, $K = 6$, $d_m = 512$, $d_{ff} = 1024$, and $r = 8$, the ratio $\frac{N_{SRP++}}{N_{iTrans}}$ is approximately 0.047, indicating a relevant small increase in model size. Figure 6 shows SRP++'s empirical running costs with varying $r$, P, and K, capturing the sum time for weight matrix adaptation and base layer forward pass. Our results confirm that the additional computational duration imposed by SRP++ is lower than 1ms, with a small increase as the hyperparameters grow. Therefore, SRP++ does not compromise the model's efficiency while effectively improving the model's performance.

## C    THEORETICAL JUSTIFICATION

**Theorem C.1** (Expressiveness Bottleneck). *Let $\bar{W} = [W\ b] \in \mathbb{R}^{T \times (L+1)}$ be the parameters in the SI's linear decoder, $Y \in \mathbb{R}^{T \times D}$ be the label sequence; the minimum attainable estimation error is*

$$\|\epsilon\|_F^2 = \sum_{t=\mathrm{rank}(\bar{W})+1}^{T} \|U_t^\top Y\|_2^2, \tag{12}$$

*where $\bar{W} = U\Sigma V^\top$ is the singular value decomposition of $\bar{W}$, $\mathrm{rank}(\bar{W}) \leq \min\{T, L+1\}$, and $\{U_i\}_{i=\mathrm{rank}+1}^{T}$ form an orthonormal basis for the null space of $\bar{W}$. Notably, this error is independent of the representation $R$ provided by encoder.*

*Proof.* Consider the least squares estimation, the aim of linear decoder is to find $\bar{R} = [R^\top, 1]^\top \in \mathbb{R}^{(L+1) \times 1}$ such that:

$$\hat{\bar{R}} = \arg\min_{\bar{R}} \|Y - \bar{W}\bar{R}\|_2^2.$$

To solve this optimization problem, we set the gradient of the cost function with respect to $\bar{R}$ to zero:

$$\frac{\partial}{\partial \bar{R}} \|Y - \bar{W}\bar{R}\|_2^2 = -2\bar{W}^\top (Y - \bar{W}\bar{R}) = 0.$$

Assuming $\bar{W}^\top \bar{W}$ is invertible, we have:

$$\bar{R} = (\bar{W}^\top \bar{W})^{-1} \bar{W}^\top Y.$$

Then, the estimation error is:

$$\epsilon = Y - \hat{Y} = \left(I - \bar{W}(\bar{W}^\top \bar{W})^{-1}\bar{W}^\top\right)Y = (I - P)Y,$$

where $P = \bar{W}(\bar{W}^\top \bar{W})^{-1}\bar{W}^\top$ is known as the projection matrix onto the column space of $\bar{W}$. Therefore, $I - P$ projects onto the orthogonal complement of the column space of $\bar{W}$.

Let the Singular Value Decomposition (SVD) of $\bar{W}$ be

$$\bar{W} = U\Sigma V^\top,$$

where $U \in \mathbb{R}^{T \times T}$ is orthogonal, $\Sigma \in \mathbb{R}^{T \times (L+1)}$ is diagonal, and $V \in \mathbb{R}^{(L+1) \times (L+1)}$ is orthogonal.

Partition $\Sigma$ and $U$ as

$$\Sigma = \begin{bmatrix} \Sigma_o & 0 \\ 0 & 0 \end{bmatrix}, \quad U = [U_o \quad U_n],$$

where $\Sigma_o \in \mathbb{R}^{o \times o}$ contains the positive singular values, $U_o \in \mathbb{R}^{T \times o}$ contains the corresponding left singular vectors, and $U_n \in \mathbb{R}^{T \times (T-o)}$ contains the left singular vectors corresponding to zero singular values, and $o = \mathrm{rank}(\bar{W})$.

The projection matrix onto the column space of $\bar{W}$ is

$$P = \bar{W}(\bar{W}^\top \bar{W})^{-1}\bar{W}^\top = U_o U_o^\top.$$

Therefore, the estimation error is

$$\epsilon = (I - P)Y = U_n U_n^\top Y.$$

The minimum attainable estimation error is

$$\|\epsilon\|_F^2 = \|U_n^\top Y\|_2^2 = \sum_{t=\mathrm{rank}(\bar{W})+1}^{T} \|U_t^\top Y\|_2^2.$$

$\square$

**Theorem C.2** (Variance Reduction of SRP++). $\mathcal{L}_{SRP++}$ *has a smaller variance than* $\mathcal{L}_{SI}$,

$$\mathrm{Var}(\mathcal{L}_{SRP++}) \leq \mathrm{Var}(\mathcal{L}_{SI}). \tag{13}$$

*Proof.* Consider the total loss expressed as the average of losses over the prediction horizons:

$$\mathcal{L} = \frac{1}{\mathrm{T}} \sum_{t=1}^{\mathrm{T}} \mathcal{L}_t,$$

where $\mathcal{L}_t$ is the MSE loss for the $t$-th prediction horizon.

The variance of the total loss is:

$$\mathrm{Var}(\mathcal{L}) = \mathrm{Var}\left(\frac{1}{\mathrm{T}} \sum_{t=1}^{\mathrm{T}} \mathcal{L}_t\right)$$

$$= \frac{1}{\mathrm{T}^2} \left(\sum_{t=1}^{\mathrm{T}} \mathrm{Var}(\mathcal{L}_t) + 2 \sum_{1 \leq t < s \leq \mathrm{T}} \mathrm{Cov}(\mathcal{L}_t, \mathcal{L}_s)\right).$$

Under modularized adaptation, the covariances between different $\mathcal{L}_t$ decrease because the LoRA modules introduce horizon-specific parameters, reducing parameter sharing. Let $\Delta \mathrm{Cov}(\mathcal{L}_t, \mathcal{L}_s) = \mathrm{Cov}_{\mathrm{SI}}(\mathcal{L}_t, \mathcal{L}_s) - \mathrm{Cov}_{\mathrm{SRP++}}(\mathcal{L}_t, \mathcal{L}_s) \geq 0$. The variance difference is then:

$$\mathrm{Var}(\mathcal{L}_{\mathrm{SI}}) - \mathrm{Var}(\mathcal{L}_{\mathrm{SRP++}}) = \frac{2}{\mathrm{T}^2} \sum_{1 \leq t < s \leq \mathrm{T}} \Delta \mathrm{Cov}(\mathcal{L}_t, \mathcal{L}_s) \geq 0.$$

Therefore,

$$\mathrm{Var}(\mathcal{L}_{\mathrm{SRP++}}) = \mathrm{Var}(\mathcal{L}_{\mathrm{SI}}) - \frac{2}{\mathrm{T}^2} \sum_{1 \leq t < s \leq \mathrm{T}} \Delta \mathrm{Cov}(\mathcal{L}_t, \mathcal{L}_s)$$

$$\leq \mathrm{Var}(\mathcal{L}_{\mathrm{SI}}).$$

The equation realize only when all the LoRA modules do not contribute to the improvement of the prediction performance with corresponding time step. However, we didn't recognize that phenomenon in our experiments.

$\square$

## D REPRODUCTION DETAILS

### D.1 DATASET DESCRIPTIONS

The datasets used in this study span a variety of domains and time resolutions, each with distinct characteristics that are well-suited for evaluating time-series forecasting:

- **ETT** (Li et al., 2021): This dataset consists of data from 7 key variables related to electricity transformers, collected between July 2016 and July 2018. It includes four subsets: ETTh1 and ETTh2, which are recorded hourly, and ETTm1 and ETTm2, which are recorded every 15 minutes.

- **ECL (Electricity Consumption Load)** (Wu et al., 2021): This dataset contains hourly electricity consumption data from 321 clients, offering insights into power usage patterns over time.

- **Traffic** (Wu et al., 2021): This dataset comprises hourly road occupancy rates collected by 862 sensors deployed across the freeways in the San Francisco Bay Area. The data spans the period from January 2015 to December 2016, reflecting traffic conditions over time.

- **Weather** (Wu et al., 2021): This dataset includes 21 meteorological variables, recorded every 10 minutes throughout 2020 at the Max Planck Biogeochemistry Institute's weather station. It provides a comprehensive set of climate-related factors for forecasting purposes.

Table 4: Detailed dataset descriptions. *D* denotes the number of variates. `Forecast Length` denotes the prediction lengths investigated in this dataset. `Frequency` denotes the sampling interval of time points. `Train, Validation, Test` denotes the number of samples employed in each split. The taxonomy and statistic are aligned with the recent works (Wu et al., 2023; Liu et al., 2024).

| Dataset | D | Forecast Length | Split Ratio | Frequency | Domain |
|---------|-----|-----------------|-------------------|-----------|----------------|
| ETTm1   | 7   | 96,192,336,720  | 34465/11521/11521 | 15mins    | Electricity    |
| ETTm2   | 7   | 96,192,336,720  | 34465/11521/11521 | 15mins    | Electricity    |
| ETTh1   | 7   | 96,192,336,720  | 8545/2881/2881    | Hourly    | Electricity    |
| ETTh2   | 7   | 96,192,336,720  | 8545/2881/2881    | Hourly    | Electricity    |
| ECL     | 321 | 96,192,336,720  | 18317/2633/5261   | Hourly    | Electricity    |
| Traffic | 862 | 96,192,336,720  | 12185/1757/3509   | Hourly    | Transportation |
| Weather | 21  | 96,192,336,720  | 36792/5271/10540  | 10mins    | Weather        |
| PEMS03  | 358 | 12,24,36,48     | 15617/5135/5135   | 5mins     | Transportation |
| PEMS08  | 170 | 12,24,36,48     | 10690/3548/265    | 5mins     | Transportation |

- **PEMS** (Liu et al., 2022b): This dataset consists of public traffic data from the California highway network, with recordings collected every 5 minutes. We utilize two subsets in this study: PEMS03 and PEMS08, which are frequently adopted in traffic forecasting benchmarks.

For all datasets, the data processing and division into training, validation, and test sets follow the protocols established by TimesNet (Wu et al., 2023) and iTransformer (Liu et al., 2024), ensuring a consistent chronological split to avoid data leakage. The standardized lookback window is set at 96 for the ETT, ECL, Traffic, Weather, and PEMS datasets. Prediction horizons vary across datasets, with forecasting lengths of 96, 192, 336, and 720 for the first five datasets, and shorter horizons of 12, 24, 36, and 48 for the PEMS subsets. Detailed specifications of each dataset can be found in Table 4.

## D.2 IMPLEMENTATION DETAILS

The baseline models in this study were carefully reproduced using training scripts from the TimesNet Repository (Wu et al., 2023), ensuring full reproducibility verification. All models were trained using the Adam optimizer (Kingma & Ba, 2015) while minimizing the MSE loss. A batch size of 32 was maintained consistently across all experiments. Training was performed for a maximum of 10 epochs, with an early stopping criterion triggered if no improvement in validation performance was observed for 3 consecutive epochs.

For the experiments integrating SRP++ into existing forecasting models, we strictly adhered to the original hyperparameter settings as outlined in the respective publications during the pre-training phase. The forecasting horizon of pre-trained models were set to $T/K$, where $K \in \{2, 3, 4, 6\}$. Pre-training was limited to a maximum of 5 epochs, with early stopping applied after 2 epochs without improvement. During the adaptation phase, only the rank $r$ in the LoRA expert module, the learning rate $\eta$ and the number of LoRA expert modules P were tuned, as these parameters are crucial for adjusting the differences in weight magnitudes between the base models and the LoRA expert modules. Specifically, we tuned $r$ in $\{4, 8, 16, 32, 64\}$, P in $\{2, 4, 6, 8, 10\}$, and $\eta$ in $\{2 \times 10^{-4}, 5 \times 10^{-4}, 10^{-3}, 2 \times 10^{-3}, 5 \times 10^{-3}\}$. Adaptation was performed to minimize the MSE averaged over all prediction lengths, with forecasting segments selected from range $\{2, 3, 4, 6\}$. The current results sufficiently demonstrate the effectiveness of SRP++, showing that its efficacy is not dependent on highly specific hyperparameter configurations.

Table 5: Full results on the multi-step forecasting task. The length of history window is set to 96 for all baselines. `Avg` indicates the results averaged over forecasting lengths: T=96, 192, 336 and 720.

| Models | | SRP++ (Ours) | | iTransformer (2024) | | FreTS (2023) | | TimesNet (2023) | | TiDE (2023) | | DLinear (2023) | | FEDformer (2022) | | Autoformer (2021) | | Informer (2021) | | Transformer (2017) | | TCN (2017) | |
|---|---|---|---|---|---|---|---|---|---|---|---|---|---|---|---|---|---|---|---|---|---|---|---|
| Metrics | | MSE | MAE | MSE | MAE | MSE | MAE | MSE | MAE | MSE | MAE | MSE | MAE | MSE | MAE | MSE | MAE | MSE | MAE | MSE | MAE | MSE | MAE |
| ETTm1 | 96 | 0.330 | 0.364 | 0.346 | 0.379 | 0.339 | 0.374 | 0.338 | 0.379 | 0.364 | 0.387 | 0.345 | 0.372 | 0.389 | 0.427 | 0.468 | 0.463 | 0.633 | 0.560 | 0.591 | 0.549 | 0.887 | 0.613 |
| | 192 | 0.378 | 0.392 | 0.392 | 0.400 | 0.382 | 0.397 | 0.389 | 0.400 | 0.398 | 0.404 | 0.381 | 0.390 | 0.402 | 0.431 | 0.573 | 0.509 | 0.736 | 0.625 | 0.704 | 0.629 | 0.877 | 0.626 |
| | 336 | 0.414 | 0.416 | 0.427 | 0.422 | 0.421 | 0.426 | 0.429 | 0.428 | 0.428 | 0.425 | 0.414 | 0.414 | 0.438 | 0.451 | 0.596 | 0.527 | 1.061 | 0.787 | 1.171 | 0.861 | 0.890 | 0.636 |
| | 720 | 0.480 | 0.453 | 0.494 | 0.461 | 0.485 | 0.462 | 0.495 | 0.464 | 0.487 | 0.461 | 0.473 | 0.451 | 0.529 | 0.498 | 0.749 | 0.569 | 1.119 | 0.801 | 1.307 | 0.893 | 0.911 | 0.653 |
| | Avg | 0.400 | 0.406 | 0.415 | 0.416 | 0.407 | 0.415 | 0.413 | 0.418 | 0.419 | 0.419 | 0.404 | 0.407 | 0.440 | 0.451 | 0.596 | 0.517 | 0.887 | 0.693 | 0.943 | 0.733 | 0.891 | 0.632 |
| ETTm2 | 96 | 0.181 | 0.262 | 0.184 | 0.266 | 0.190 | 0.282 | 0.185 | 0.264 | 0.207 | 0.305 | 0.195 | 0.294 | 0.194 | 0.284 | 0.240 | 0.319 | 0.541 | 0.581 | 0.317 | 0.408 | 3.125 | 1.345 |
| | 192 | 0.247 | 0.307 | 0.257 | 0.315 | 0.260 | 0.329 | 0.254 | 0.307 | 0.290 | 0.364 | 0.283 | 0.359 | 0.264 | 0.324 | 0.300 | 0.349 | 0.527 | 0.558 | 1.069 | 0.758 | 3.130 | 1.350 |
| | 336 | 0.312 | 0.347 | 0.315 | 0.351 | 0.373 | 0.405 | 0.314 | 0.345 | 0.377 | 0.422 | 0.384 | 0.427 | 0.319 | 0.359 | 0.339 | 0.375 | 1.126 | 0.797 | 1.325 | 0.869 | 3.185 | 1.375 |
| | 720 | 0.407 | 0.403 | 0.419 | 0.409 | 0.517 | 0.499 | 0.414 | 0.413 | 0.558 | 0.524 | 0.516 | 0.502 | 0.430 | 0.424 | 0.423 | 0.421 | 2.828 | 1.268 | 2.576 | 1.223 | 4.203 | 1.658 |
| | Avg | 0.287 | 0.330 | 0.294 | 0.335 | 0.335 | 0.379 | 0.297 | 0.332 | 0.358 | 0.404 | 0.344 | 0.396 | 0.302 | 0.348 | 0.326 | 0.366 | 1.256 | 0.801 | 1.322 | 0.814 | 3.411 | 1.432 |
| ETTh1 | 96 | 0.379 | 0.400 | 0.390 | 0.410 | 0.399 | 0.412 | 0.422 | 0.433 | 0.479 | 0.464 | 0.396 | 0.410 | 0.377 | 0.418 | 0.423 | 0.441 | 0.920 | 0.745 | 0.796 | 0.691 | 0.767 | 0.633 |
| | 192 | 0.436 | 0.432 | 0.443 | 0.441 | 0.453 | 0.443 | 0.465 | 0.457 | 0.521 | 0.503 | 0.449 | 0.444 | 0.421 | 0.445 | 0.498 | 0.485 | 0.998 | 0.781 | 0.813 | 0.699 | 0.739 | 0.619 |
| | 336 | 0.477 | 0.456 | 0.480 | 0.457 | 0.503 | 0.475 | 0.492 | 0.470 | 0.659 | 0.603 | 0.487 | 0.465 | 0.468 | 0.472 | 0.506 | 0.496 | 1.091 | 0.812 | 1.181 | 0.876 | 0.717 | 0.613 |
| | 720 | 0.480 | 0.478 | 0.484 | 0.479 | 0.596 | 0.565 | 0.532 | 0.502 | 0.893 | 0.736 | 0.516 | 0.513 | 0.500 | 0.493 | 0.477 | 0.487 | 1.247 | 0.887 | 1.182 | 0.885 | 0.828 | 0.678 |
| | Avg | 0.443 | 0.441 | 0.449 | 0.447 | 0.488 | 0.474 | 0.478 | 0.466 | 0.628 | 0.574 | 0.462 | 0.458 | 0.441 | 0.457 | 0.476 | 0.477 | 1.064 | 0.806 | 0.993 | 0.788 | 0.763 | 0.636 |
| ETTh2 | 96 | 0.293 | 0.341 | 0.301 | 0.349 | 0.350 | 0.403 | 0.320 | 0.364 | 0.400 | 0.440 | 0.343 | 0.396 | 0.347 | 0.391 | 0.383 | 0.424 | 2.340 | 1.220 | 2.072 | 1.140 | 3.171 | 1.364 |
| | 192 | 0.377 | 0.397 | 0.382 | 0.402 | 0.472 | 0.475 | 0.409 | 0.417 | 0.528 | 0.509 | 0.473 | 0.474 | 0.430 | 0.443 | 0.557 | 0.511 | 6.284 | 2.078 | 5.081 | 1.814 | 3.222 | 1.398 |
| | 336 | 0.421 | 0.429 | 0.430 | 0.434 | 0.564 | 0.528 | 0.449 | 0.451 | 0.643 | 0.571 | 0.603 | 0.546 | 0.469 | 0.475 | 0.470 | 0.481 | 4.824 | 1.853 | 3.564 | 1.475 | 3.306 | 1.452 |
| | 720 | 0.418 | 0.438 | 0.447 | 0.455 | 0.815 | 0.654 | 0.473 | 0.474 | 0.874 | 0.679 | 0.812 | 0.650 | 0.473 | 0.480 | 0.501 | 0.515 | 3.985 | 1.724 | 2.469 | 1.247 | 3.599 | 1.565 |
| | Avg | 0.377 | 0.401 | 0.390 | 0.410 | 0.550 | 0.515 | 0.413 | 0.426 | 0.611 | 0.550 | 0.558 | 0.516 | 0.430 | 0.447 | 0.478 | 0.483 | 4.358 | 1.719 | 3.296 | 1.419 | 3.325 | 1.445 |
| ECL | 96 | 0.147 | 0.237 | 0.148 | 0.239 | 0.189 | 0.277 | 0.171 | 0.273 | 0.237 | 0.329 | 0.210 | 0.302 | 0.200 | 0.315 | 0.199 | 0.315 | 0.315 | 0.398 | 0.252 | 0.352 | 0.688 | 0.621 |
| | 192 | 0.163 | 0.254 | 0.167 | 0.258 | 0.193 | 0.282 | 0.188 | 0.289 | 0.236 | 0.330 | 0.210 | 0.305 | 0.207 | 0.322 | 0.215 | 0.327 | 0.327 | 0.411 | 0.266 | 0.364 | 0.587 | 0.582 |
| | 336 | 0.179 | 0.270 | 0.179 | 0.272 | 0.207 | 0.296 | 0.208 | 0.304 | 0.249 | 0.344 | 0.223 | 0.319 | 0.226 | 0.340 | 0.232 | 0.343 | 0.354 | 0.434 | 0.292 | 0.383 | 0.590 | 0.588 |
| | 720 | 0.208 | 0.297 | 0.209 | 0.298 | 0.245 | 0.332 | 0.289 | 0.363 | 0.284 | 0.373 | 0.258 | 0.350 | 0.282 | 0.379 | 0.268 | 0.371 | 0.343 | 0.423 | 0.287 | 0.371 | 0.602 | 0.601 |
| | Avg | 0.174 | 0.265 | 0.176 | 0.267 | 0.209 | 0.297 | 0.214 | 0.307 | 0.251 | 0.344 | 0.225 | 0.319 | 0.229 | 0.339 | 0.228 | 0.339 | 0.335 | 0.416 | 0.274 | 0.367 | 0.617 | 0.598 |
| Traffic | 96 | 0.396 | 0.271 | 0.397 | 0.272 | 0.528 | 0.341 | 0.504 | 0.298 | 0.805 | 0.493 | 0.697 | 0.429 | 0.577 | 0.362 | 0.609 | 0.385 | 0.698 | 0.390 | 0.686 | 0.385 | 1.451 | 0.744 |
| | 192 | 0.416 | 0.278 | 0.418 | 0.279 | 0.531 | 0.338 | 0.526 | 0.305 | 0.756 | 0.474 | 0.647 | 0.407 | 0.603 | 0.372 | 0.633 | 0.398 | 0.697 | 0.386 | 0.679 | 0.377 | 0.842 | 0.622 |
| | 336 | 0.425 | 0.283 | 0.432 | 0.286 | 0.551 | 0.345 | 0.540 | 0.310 | 0.762 | 0.477 | 0.653 | 0.410 | 0.615 | 0.378 | 0.637 | 0.398 | 0.715 | 0.397 | 0.663 | 0.361 | 0.844 | 0.620 |
| | 720 | 0.464 | 0.304 | 0.467 | 0.305 | 0.598 | 0.367 | 0.570 | 0.324 | 0.719 | 0.449 | 0.694 | 0.429 | 0.649 | 0.403 | 0.668 | 0.415 | 0.797 | 0.443 | 0.693 | 0.381 | 0.867 | 0.624 |
| | Avg | 0.425 | 0.284 | 0.428 | 0.286 | 0.552 | 0.348 | 0.535 | 0.309 | 0.760 | 0.473 | 0.673 | 0.419 | 0.611 | 0.379 | 0.637 | 0.399 | 0.727 | 0.404 | 0.680 | 0.376 | 1.001 | 0.652 |
| Weather | 96 | 0.173 | 0.211 | 0.201 | 0.247 | 0.184 | 0.239 | 0.178 | 0.226 | 0.202 | 0.261 | 0.197 | 0.259 | 0.221 | 0.304 | 0.284 | 0.355 | 0.383 | 0.438 | 0.332 | 0.383 | 0.610 | 0.568 |
| | 192 | 0.246 | 0.280 | 0.250 | 0.283 | 0.223 | 0.275 | 0.227 | 0.266 | 0.242 | 0.298 | 0.236 | 0.294 | 0.275 | 0.345 | 0.313 | 0.371 | 0.415 | 0.449 | 0.634 | 0.539 | 0.541 | 0.552 |
| | 336 | 0.277 | 0.296 | 0.302 | 0.317 | 0.272 | 0.316 | 0.283 | 0.305 | 0.287 | 0.335 | 0.282 | 0.332 | 0.338 | 0.379 | 0.359 | 0.393 | 0.618 | 0.551 | 0.656 | 0.579 | 0.565 | 0.569 |
| | 720 | 0.367 | 0.356 | 0.370 | 0.362 | 0.340 | 0.363 | 0.359 | 0.355 | 0.351 | 0.386 | 0.347 | 0.384 | 0.408 | 0.418 | 0.440 | 0.446 | 0.963 | 0.726 | 0.908 | 0.706 | 0.622 | 0.601 |
| | Avg | 0.266 | 0.286 | 0.281 | 0.302 | 0.255 | 0.299 | 0.262 | 0.288 | 0.271 | 0.320 | 0.265 | 0.317 | 0.311 | 0.361 | 0.349 | 0.391 | 0.595 | 0.541 | 0.632 | 0.552 | 0.584 | 0.572 |
| PEMS03 | 12 | 0.067 | 0.173 | 0.069 | 0.175 | 0.083 | 0.194 | 0.082 | 0.188 | 0.117 | 0.225 | 0.122 | 0.245 | 0.123 | 0.248 | 0.239 | 0.365 | 0.122 | 0.226 | 0.107 | 0.209 | 0.632 | 0.606 |
| | 24 | 0.095 | 0.205 | 0.098 | 0.210 | 0.127 | 0.241 | 0.110 | 0.216 | 0.233 | 0.320 | 0.202 | 0.320 | 0.160 | 0.287 | 0.492 | 0.506 | 0.129 | 0.233 | 0.121 | 0.227 | 0.655 | 0.626 |
| | 36 | 0.127 | 0.240 | 0.131 | 0.243 | 0.169 | 0.281 | 0.133 | 0.236 | 0.380 | 0.422 | 0.275 | 0.382 | 0.191 | 0.321 | 0.399 | 0.459 | 0.143 | 0.249 | 0.133 | 0.243 | 0.678 | 0.644 |
| | 48 | 0.159 | 0.270 | 0.164 | 0.275 | 0.204 | 0.311 | 0.146 | 0.251 | 0.536 | 0.511 | 0.335 | 0.429 | 0.223 | 0.350 | 0.875 | 0.723 | 0.153 | 0.255 | 0.144 | 0.253 | 0.699 | 0.659 |
| | Avg | 0.112 | 0.222 | 0.116 | 0.226 | 0.146 | 0.257 | 0.118 | 0.223 | 0.316 | 0.370 | 0.233 | 0.344 | 0.174 | 0.302 | 0.501 | 0.513 | 0.137 | 0.241 | 0.126 | 0.233 | 0.666 | 0.634 |
| PEMS08 | 12 | 0.079 | 0.181 | 0.085 | 0.189 | 0.095 | 0.204 | 0.110 | 0.209 | 0.121 | 0.231 | 0.152 | 0.274 | 0.175 | 0.275 | 0.446 | 0.483 | 0.268 | 0.281 | 0.213 | 0.236 | 0.680 | 0.607 |
| | 24 | 0.114 | 0.218 | 0.131 | 0.236 | 0.150 | 0.259 | 0.142 | 0.239 | 0.232 | 0.326 | 0.245 | 0.350 | 0.211 | 0.305 | 0.488 | 0.509 | 0.296 | 0.302 | 0.238 | 0.256 | 0.701 | 0.622 |
| | 36 | 0.158 | 0.256 | 0.182 | 0.282 | 0.202 | 0.305 | 0.167 | 0.258 | 0.379 | 0.428 | 0.344 | 0.417 | 0.250 | 0.338 | 0.532 | 0.513 | 0.340 | 0.327 | 0.263 | 0.277 | 0.727 | 0.637 |
| | 48 | 0.203 | 0.290 | 0.236 | 0.323 | 0.250 | 0.341 | 0.195 | 0.274 | 0.543 | 0.527 | 0.437 | 0.469 | 0.293 | 0.371 | 1.052 | 0.781 | 0.373 | 0.345 | 0.283 | 0.295 | 0.746 | 0.648 |
| | Avg | 0.138 | 0.236 | 0.159 | 0.258 | 0.174 | 0.277 | 0.154 | 0.245 | 0.319 | 0.378 | 0.294 | 0.377 | 0.232 | 0.322 | 0.630 | 0.572 | 0.319 | 0.314 | 0.249 | 0.266 | 0.713 | 0.629 |
| 1st Count | | 33 | 36 | 0 | 0 | 4 | 0 | 2 | 6 | 0 | 0 | 1 | 3 | 4 | 0 | 1 | 0 | 0 | 0 | 0 | 0 | 0 | 0 |

# E MORE EXPERIMENTAL RESULTS

## E.1 OVERALL PERFORMANCE

We present a comprehensive comparison of the multi-step forecasting task in Table 5. The iTransformer model serves as the base model for implementing the SRP++ framework. Despite iTransformer's initial performance gap compared to other state-of-the-art baseline models, integrating SRP++ significantly enhances its forecasting accuracy.

Specifically, SRP++ achieves the lowest MSE in 33 out of 45 cases and the lowest MAE in 36 out of 45 cases, illustrating its effectiveness in enhancing model performance. The improvements brought by SRP++ are particularly noticeable on challenging datasets such as ETTm1 and Traffic, where capturing long-term dependencies is crucial. These results underscore the robustness and adaptability of the SRP++ framework. While there are a few instances where SRP++ does not achieve the top performance, this can be attributed to the inherent advantages of specific models in particular contexts. For example, FreTS shows competitive results on the Weather dataset, where its architectural design may better suit certain meteorological patterns. Nonetheless, the overall performance of SRP++ demonstrates its strength as a adaptation framework for time-series forecasting, consis-

tently mitigating the expressiveness bottleneck and delivering superior results across a wide range of forecasting tasks.

## E.2 GENERALIZATION STUDIES

We visualize the three prominent PEFT methods: Adapter, LoRA and IA$^3$ in Figure 7, showing the architectural modifications introduced by each PEFT technique within a typical transformer block. Adapter, shown in the upper left, introduces additional fully connected (FC) layers and a short-cut connection after the FFN layer. The Adapter module typically consists of a down-projection FC layer, followed by a non-linearity (often ReLU), an up-projection FC layer, and a residual connection. This approach provides a compact, trainable module that can adjust the model's behavior for specific tasks without modifying the entire network. LoRA, illustrated on the lower left, modifies the FFN layer by adding low-rank matrices to the frozen weight matrix. LoRA decomposes the weight update into two low-rank matrices, enabling the model

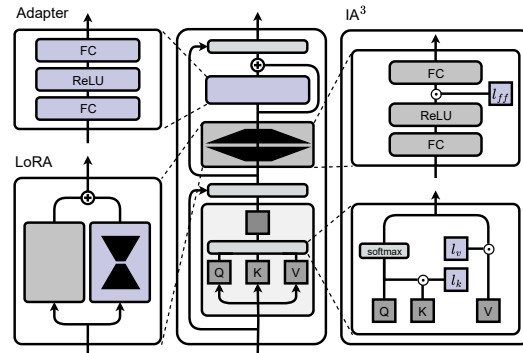

Figure 7: Visualization of common parameter-efficient adaptation strategies.

to learn task-specific adaptations in a parameter-efficient manner. IA$^3$, shown on the right, works by applying learnable scaling factors to the key and value projections in the self-attention mechanism and to the hidden representations in the feed-forward network, allowing for fine-grained control over the model's behavior with minimal additional parameters.

For a fair comparison, we matched the low-rank configuration of the Adapter and the LoRA modules with that of the LoRA expert in SRP++, applying both to the feed-forward network (FFN) layers. IA$^3$, on the other hand, dynamically adjusts the weights of intermediate hidden vectors in both the FFN and attention layers.

## E.3 HYPERPARAMETER SENSITIVITY

In this section, we further analyze the sensitivity of SRP++ to three key hyperparameters under different segment number settings: the low-rank value $r$ in the LoRA modules, the learning rate $\eta$, and the number of LoRA expert modules P. The experiments are conducted using iTransformer across two datasets, ETTh1 and Weather, with the results visualized in Figure 8 and Figure 9.

The results generally indicate that both extremely low and high values of rank $r$ and learning rate $\eta$ lead to performance degradation. This pattern suggests that overly high ranks may lead to overfitting, while excessively low ranks may not provide sufficient flexibility for effective model adaptation. Similarly, very high learning rates can cause instability in training, while very low learning rates may result in slow convergence or getting stuck in suboptimal solutions. Interestingly, for the number of expert modules P, SRP++ exhibits high stability as long as P is not significantly lower than the number of segments K. This robustness indicates that the model can effectively leverage multiple experts to capture diverse patterns across different forecasting segments.

These findings indicate that adaptation the hyperparameters for each specific forecasting horizon can further improve performance. However, our results demonstrate that even without exhaustive tuning, the SRP++ framework delivers robust performance improvements, highlighting its flexibility and effectiveness in time-series forecasting tasks.

## E.4 EXTRA DISCUSSIONS

**Discussion on adaptation bias.**    In this section, we extend our analysis by evaluating the impact of jointly adaptation both the weight and bias within the LoRA modules under the SRP++ framework. The results, detailed in Table 6, cover a variety of datasets.

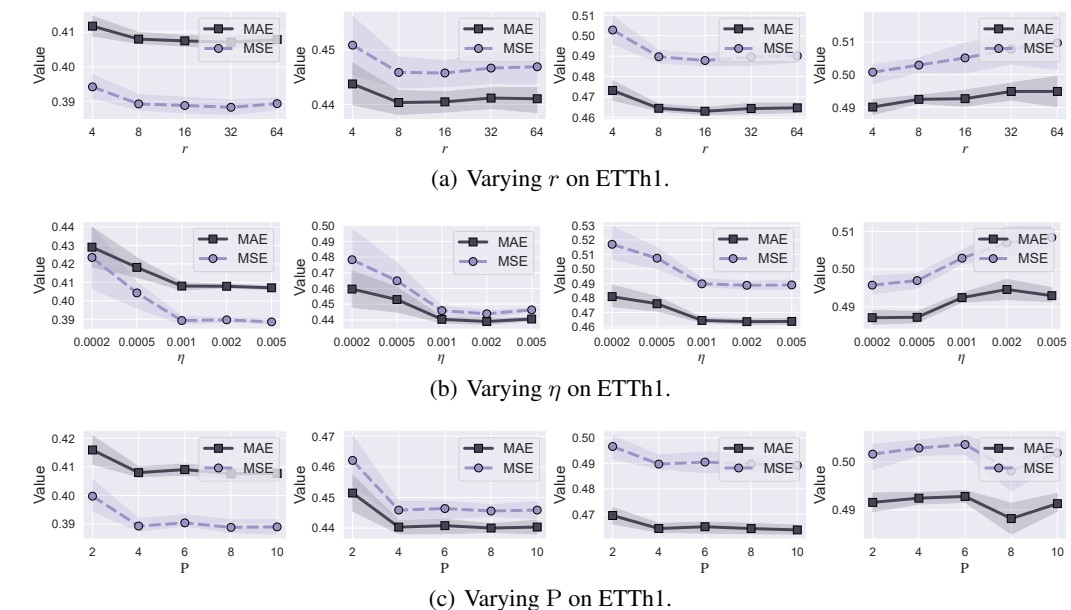

Figure 8: Performance of iTransformer enhanced by SRP++ given different low rank of LoRA modules $r$, learning rate $\eta$ and the number of experts P. Different columns correspond to different number of forecasting length T (from left to right: 96, 192, 336, 720). The results are averaged on four forecasting segment number (2, 3, 4, 6) with shaded areas being 50% confidence intervals.

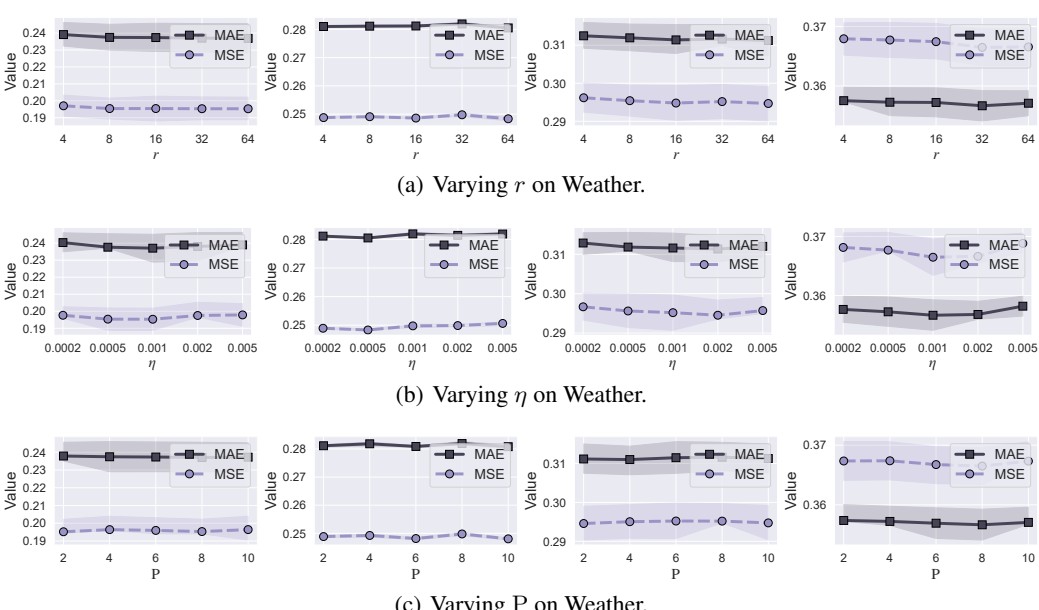

Figure 9: Performance of iTransformer enhanced by SRP++ given different low rank of LoRA modules $r$, learning rate $\eta$ and the number of experts P. Different columns correspond to different number of forecasting length T (from left to right: 96, 192, 336, 720). The results are averaged on four forecasting segment number (2, 3, 4, 6) with shaded areas being 50% confidence intervals.

Overall, the findings indicate that the joint adaptation of weights and biases (SRP++/WB) results in performance degradation compared to adaptation weights alone under the SRP++ framework. Specifically, SRP++/WB tends to increase the risk of overfitting, particularly in datasets with complex temporal dependencies, where the added non-low-rank parameters introduce excessive flexibil-

Table 6: Full results on the multi-step forecasting task with jointly adaptation weight and bias under SRP++ framework. The length of history window is set to 96 for all baselines. *Avg* indicates the results averaged over forecasting lengths: T=96, 192, 336 and 720 for ETT, ECL, Traffic and Weather dataset, T=12,24,36 and 48 for PEMS datasets.

| Datasets | ETTm1 | | ETTm2 | | ETTh1 | | ETTh2 | | ECL | | Traffic | | Weather | | | PEMS03 | | PEMS08 | |
|---|---|---|---|---|---|---|---|---|---|---|---|---|---|---|---|---|---|---|---|
| Metrics | MSE | MAE | MSE | MAE | MSE | MAE | MSE | MAE | MSE | MAE | MSE | MAE | MSE | MAE | | MSE | MAE | MSE | MAE |
| SRP++/WB 96 | 0.337 | 0.368 | 0.182 | 0.263 | 0.378 | 0.400 | 0.297 | 0.345 | 0.147 | 0.237 | 0.394 | 0.270 | 0.201 | 0.245 | 12 | 0.070 | 0.175 | 0.079 | 0.181 |
| 192 | 0.382 | 0.394 | 0.246 | 0.306 | 0.434 | 0.431 | 0.379 | 0.398 | 0.163 | 0.254 | 0.420 | 0.282 | 0.246 | 0.279 | 24 | 0.095 | 0.205 | 0.114 | 0.218 |
| 336 | 0.419 | 0.420 | 0.315 | 0.350 | 0.477 | 0.456 | 0.420 | 0.431 | 0.182 | 0.273 | 0.640 | 0.426 | 0.280 | 0.297 | 36 | 0.127 | 0.239 | 0.158 | 0.256 |
| 720 | 0.483 | 0.456 | 0.415 | 0.407 | 0.495 | 0.488 | 0.432 | 0.448 | 0.211 | 0.300 | 0.585 | 0.352 | 0.367 | 0.356 | 48 | 0.158 | 0.269 | 0.202 | 0.290 |
| Avg | 0.405 | 0.409 | 0.289 | 0.332 | 0.446 | 0.444 | 0.382 | 0.405 | 0.176 | 0.266 | 0.549 | 0.353 | 0.274 | 0.294 | Avg | 0.112 | 0.222 | 0.138 | 0.236 |

ity, making the model more difficult to optimize effectively. This is especially evident in the larger forecasting horizons, where the performance gap becomes more pronounced.

However, SRP++/WB generally improves the performance of the SI paradigm, suggesting that while the joint adaptation strategy introduces challenges for the SRP++ framework, it may still offer some benefit in simpler or less modular adaptation strategies. The overall results affirm that focusing on weight adaptation alone is a more effective approach for leveraging the full potential of SRP++, ensuring better generalization and predictive accuracy across diverse forecasting tasks.

**Discussion on Layer-wise Adaptation.** In this section, we explore the impact of layer-wise adaptation in the multi-layer iTransformer model across various datasets. Specifically, for the ETT dataset, we adaptd the first (SRP++/L1) and second (SRP++/L2) layers, while for the ECL, Traffic, Weather, and PEMS datasets, we extended the study to include the third layer (SRP++/L3). The results are detailed in Tables 7 and Table 8.

From the experiments, we observe that layer-wise adaptation generally yields positive performance improvements over the SI paradigm in most cases, particularly for datasets like ETTm1, ETTm2, Weather, and ECL. In these datasets, adaptation a single layer was sufficient to capture the temporal dependencies effectively, leading to reduced MSE and MAE. For instance, in the ETT datasets, both SRP++/L1 and SRP++/L2 show competitive results compared to full-layer adaptation, indicating that focusing on specific layers can provide significant computational savings without sacrificing accuracy.

However, for more complex datasets such as Traffic and PEMS03, adaptation a single layer did not achieve results better than those obtained with the SI paradigm. This could be due to the disruption of key inter-layer interactions that are crucial for the hierarchical processing in Transformer-based models. These interactions are particularly important in datasets with complex temporal patterns or multiple variates, where adjustments in a single layer may not provide enough capacity to adapt to the nuances of the time-series data.

The results suggest that while layer-wise adaptation can be beneficial in reducing the computational overhead and maintaining high performance, it is dataset-dependent. In datasets with more complex structures, a more comprehensive adaptation strategy involving multiple layers or full-layer adaptation may be necessary to avoid underfitting and fully capture the temporal dependencies in the data.

# F STATEMENT ON THE USE OF LARGE LANGUAGE MODELS (LLMS)

We used LLM-based tools solely as copy-editing assistants to improve grammar, spelling, and readability of text written by the authors. The tools were not used for research ideation, literature review, technical content generation, data analysis, result generation, or figure creation. All scientific content was conceived and written by the authors. Suggestions from the tools were limited to surface-level language polishing and were manually reviewed to ensure that meaning and technical correctness were preserved.

Table 7: Full results on the multi-step forecasting task with layer-wise adaptation under SRP++ framework.

| Datasets | | ETTm1 | | ETTm2 | | ETTh1 | | ETTh2 | |
|---|---|---|---|---|---|---|---|---|---|
| Metrics | | MSE | MAE | MSE | MAE | MSE | MAE | MSE | MAE |
| SRP++/L1 | 96 | 0.342 | 0.373 | 0.183 | 0.263 | 0.379 | 0.400 | 0.293 | 0.342 |
| | 192 | 0.386 | 0.396 | 0.251 | 0.310 | 0.436 | 0.432 | 0.380 | 0.397 |
| | 336 | 0.426 | 0.424 | 0.315 | 0.352 | 0.477 | 0.456 | 0.428 | 0.434 |
| | 720 | 0.495 | 0.462 | 0.412 | 0.406 | 0.486 | 0.481 | 0.428 | 0.445 |
| | Avg | 0.412 | 0.414 | 0.290 | 0.333 | 0.444 | 0.442 | 0.382 | 0.404 |
| SRP++/L2 | 96 | 0.335 | 0.369 | 0.181 | 0.262 | 0.377 | 0.398 | 0.296 | 0.343 |
| | 192 | 0.382 | 0.395 | 0.246 | 0.306 | 0.434 | 0.431 | 0.384 | 0.402 |
| | 336 | 0.414 | 0.418 | 0.308 | 0.344 | 0.480 | 0.459 | 0.432 | 0.438 |
| | 720 | 0.480 | 0.454 | 0.411 | 0.405 | 0.498 | 0.490 | 0.430 | 0.445 |
| | Avg | 0.403 | 0.409 | 0.286 | 0.329 | 0.447 | 0.445 | 0.385 | 0.407 |

Table 8: Full results on the multi-step forecasting task with layer-wise adaptation under SRP++ framework.

| Datasets | | ECL | | Traffic | | Weather | | | PEMS03 | | PEMS08 | |
|---|---|---|---|---|---|---|---|---|---|---|---|---|
| Metrics | | MSE | MAE | MSE | MAE | MSE | MAE | | MSE | MAE | MSE | MAE |
| SRP++/L1 | 96 | 0.149 | 0.239 | 0.401 | 0.275 | 0.203 | 0.245 | 12 | 0.069 | 0.175 | 0.081 | 0.185 |
| | 192 | 0.164 | 0.255 | 0.425 | 0.286 | 0.250 | 0.281 | 24 | 0.099 | 0.210 | 0.123 | 0.228 |
| | 336 | 0.182 | 0.274 | 0.432 | 0.286 | 0.280 | 0.298 | 36 | 0.136 | 0.249 | 0.173 | 0.271 |
| | 720 | 0.215 | 0.301 | 0.480 | 0.320 | 0.368 | 0.358 | 48 | 0.172 | 0.284 | 0.222 | 0.309 |
| | Avg | 0.177 | 0.267 | 0.435 | 0.292 | 0.275 | 0.296 | Avg | 0.119 | 0.229 | 0.150 | 0.249 |
| SRP++/L2 | 96 | 0.149 | 0.240 | 0.401 | 0.276 | 0.203 | 0.246 | 12 | 0.069 | 0.175 | 0.082 | 0.185 |
| | 192 | 0.163 | 0.254 | 0.423 | 0.285 | 0.248 | 0.280 | 24 | 0.099 | 0.209 | 0.122 | 0.228 |
| | 336 | 0.182 | 0.274 | 0.431 | 0.286 | 0.281 | 0.298 | 36 | 0.135 | 0.248 | 0.174 | 0.272 |
| | 720 | 0.212 | 0.299 | 0.479 | 0.319 | 0.369 | 0.358 | 48 | 0.171 | 0.281 | 0.218 | 0.305 |
| | Avg | 0.176 | 0.267 | 0.433 | 0.292 | 0.275 | 0.295 | Avg | 0.118 | 0.228 | 0.149 | 0.247 |
| SRP++/L3 | 96 | 0.150 | 0.240 | 0.402 | 0.276 | 0.201 | 0.245 | 12 | 0.069 | 0.176 | 0.083 | 0.186 |
| | 192 | 0.162 | 0.254 | 0.424 | 0.287 | 0.249 | 0.280 | 24 | 0.099 | 0.210 | 0.125 | 0.230 |
| | 336 | 0.180 | 0.272 | 0.431 | 0.286 | 0.280 | 0.297 | 36 | 0.136 | 0.249 | 0.175 | 0.272 |
| | 720 | 0.210 | 0.297 | 0.481 | 0.319 | 0.369 | 0.358 | 48 | 0.171 | 0.282 | 0.219 | 0.305 |
| | Avg | 0.176 | 0.266 | 0.435 | 0.292 | 0.275 | 0.295 | Avg | 0.119 | 0.229 | 0.150 | 0.248 |

