# OpenReview forum: "On the Necessity of Step-Specific Representation Learning for Multi-Step Forecasting"
_ICLR.cc/2026/Conference — Submitted to ICLR 2026_

### Official Review · Reviewer_xyJu · 2025-11-01

**Soundness:** 2
**Presentation:** 3
**Contribution:** 2
**Rating:** 4
**Confidence:** 3

**Summary:**

The paper argues that a single step-invariant representation with a linear multi-output head is inherently rank-limited for multi-horizon forecasting, then proposes SRP/SRP++ to inject step-specificity via lightweight adapters (LoRA, segment/MoE sharing).

**Strengths:**

- Pinpoints a concrete architectural bottleneck (shared rep + linear head) and connects it to an error lower bound.
- Step-specific adapters are simple, modular, and easy to bolt onto popular backbones.

**Weaknesses:**

- The “bottleneck” proof assumes a linear multi-output head tied to a single shared representation with output dimension constraints (e.g., error > 0 when L+1<T). It does not rule out non-linear/attention decoders or architectures that already produce step-conditioned features (causal decoders, per-step cross-attention), so the claimed “unavoidable error” may not generalize to modern designs.
- Reported gains can be confounded with extra capacity/tuning from segmentation and MoE sharing; improvements are small and dataset-sensitive, making it hard to credit “step-specificity” rather than added parameters.

**Questions:**

- If the linear head is replaced by a non-linear per-step decoder (e.g., cross-attention into encoder states) or the encoder is explicitly step-conditioned, does the lower bound still bite?
- Can you provide a capacity-matched step-invariant baseline (same params/updates/latency) to isolate step-specificity from pure capacity gains?
- How does SRP++ compare to simply widening the head or using a shallow per-step MLP without segmentation?

---

### Official Review · Reviewer_9bce · 2025-11-04

**Soundness:** 1
**Presentation:** 2
**Contribution:** 1
**Rating:** 2
**Confidence:** 4

**Summary:**

The paper argues that common direct multi-step forecasters use a step-invariant (SI) latent representation (one shared vector decoded to all horizons), which induces a fundamental expressiveness bottleneck. Theoretically, with an SI linear head, the minimum error is bounded away from zero, independent of how strong the encoder is. Proposed method, SRP: pretrain a one-step model, then for each step, freeze the backbone and attach LoRA adapters to encoder layers to produce step-specific latents (single-output head per step), eliminating the SI constraint. Extended SRP++: segment the horizon into K contiguous chunks of size and share P LoRA experts with learnable mixture weights per segment to cut cost and capture inter-step dependencies.

**Strengths:**

- Reproducibility: code is shared

**Weaknesses:**

- Motivation is not aligned with benchmarking
- Long-horizon benchmarks have limited practical significance
- Narrow architecture class described in equation (1), limiting the applicability of proposed method
- Theoretical analysis is trivial, as it considers only the case of linear encoder. As such, it is not applicable to most modern encoders, which are non-linear
- Incremental novelty: the contribution is based mostly on reusing the LoRA paradigm

**Questions:**

> real-world applications, such as finance (e.g., stock selection (Hu et al., 2025)), meteorology (e.g., weather forecast (Allen et al., 2025)), and manufacturing (e.g., process monitoring (Wang et al., 2023a)).
- The paper motivates research in several domains, as shown above. However, there is clear disconnect between this motivation, because benchmarks include weather, electricity and traffic. Most datasets used in the study are tiny and overused in the literature, raising concerns both about statistical significance of results and overfitting of the proposed architectural solution to the benchmarks. Could you please include results on larger scale benchmarks such as M4 and GiftEval, as well as M3 and TOURISM?

- I invite authors to think about practical significance of long-horison forecasting. For example, what are the contexts in which we need 700-step forward forecasts? As an example, suppose we have daily grain forecasting problem. What is the context in which we would need a forecast 1000 days forward, like 2.5 years, at daily grain? For all practical purposes, in this scenario we will probably only need 3 point forecast (3 years forward) at yearly grain or 36 months forward at monthly grain. Similar comments apply to other scenarios in which hierarchical forecasting is used to take care of long horizon planning decisions.

- The formulation of encoder-decoder framework in (1) targets a very specific underlying architecture, with constraint of encoder output in the space $\mathbb{R}^{L\times D}$. I am not at all convinced that most modern architectures can be cast in this formulation. This drastically limits the significance and impact of proposed approach.

- The proof of Theorem 3.1 assumes that the encoder R is a linear layer. No wonder that the resulting error in Theorem 3.1 does not depend on R. Moreover, the statement *Notably, this error is independent of the representation R provided by encoder* is false. Indeed, the results of the Theorem only hold if R is linear. The analysis in Theorem 3.1 is not applicable to non-linear encoders.

- The Case Study in Fig. 1 considers 2-layer MLP trained per step, and showing different representations per step. However, a sufficiently deep MLP supplied with proper position encoding and co-trained across all steps will show the same variation in representations per step. Again, this will demonstrate the fact that Theorem 3.1 does not describe how non-linear encoders operate.

- Sections 3.2 and 3.3 assume non-linear encoder, making the theory in Section 3.1 detached from the proposed implementation, and as such - irrelevant

---

> ### Comment · Reviewer_9bce · 2025-11-28
> **No author response**
>
> The authors did not provide a response. I keep my score.

---

### Official Review · Reviewer_1WRS · 2025-11-06

**Soundness:** 3
**Presentation:** 2
**Contribution:** 2
**Rating:** 4
**Confidence:** 4

**Summary:**

This work investigates the inherent forecasting errors incurred by models that rely on step-invariant time series representations, of which many of the recent transformer variants are examples. To address this limitation, the authors propose training models for stepwise predictions first, then adapting them for different forecast windows using low rank adapters. Further, the method is extended to model correlations in the forecast via a mixture of LoRA experts.

**Strengths:**

1. The problem is valid and well-posed: creating forecasts from step-invariant representations leads to an inherent limitation in expressiveness.
2. The experimental evidence suggests at least comparable performance with common neural architectures and linear models.
3. The paper is thorough in the discussion: adapter schemes, hyperparameter ablations, backbones, etc.

**Weaknesses:**

- The presentation of the paper could be improved. For instance:
	- The setup and assumptions for the main theoretical result (Theorem 3.1) could be spelled out more explicitly. Furthermore, no discussion on the result and its consequences is provided. Finally, the proof of the theorem is not referred to in the main text. I assume it's Theorem C.2 in the appendix, but that is also not correctly labeled.
	- The central claim is around step invariant representations being insufficient for long horizons. Therefore, the effect of the window length should play a major role. I would have liked to see a more thorough treatment of the relationship between window length and forecast error in the proposed method compared to some baseline. In my opinion, presenting averaged results over prediction lengths unnecessarily obscures this.
- The results presented in Table 2 are not very strong in my opinion. In 5/10 cases, the difference between SI and SRP/SRP++ is less than 3%. Even the case that the authors highlight in Section 4.2 (vs. iTransformer on ETTM1) is a 2.4% difference.
- Given that the confidence intervals in Figure 4 are relatively large, it is difficult to tell whether the reported improvements are significant.

**Questions:**

Overall, I think this paper proposes an interesting idea that could have a stronger presentation if some empirical aspects are addressed.

- Claim in Section 3.1 "This assumption becomes problematic when different forecast steps require fundamentally different representational features, particularly in long-term forecast scenarios  where $T$  is large".  Could the authors provide evidence that step invariant representations degrade at a faster rate than SRP/SRP++ as the horizon length increases?
 - In Figure 5: are the error bars on the second row of plots missing or are they just very small?
 - An implication of Theorem 3.1 is that  the error should vanish as $L$  grows. How does this transition looks in the synthetic experiment?
 - In practice, does it make a significant difference to train the base model and then the LoRA adapters vs. training the whole architecture jointly?

---

### Official Review · Reviewer_r5mT · 2025-11-06

**Soundness:** 3
**Presentation:** 3
**Contribution:** 2
**Rating:** 4
**Confidence:** 3

**Summary:**

The paper studies multi-step direct time-series forecasting with encoder/linear decoder architectures where a single, step-invariant representation is used to predict all future steps. It shows theoretically that, with a linear decoder, this shared representation creates an expressiveness bottleneck for long horizons, leading to irreducible approximation error. To mitigate this, the authors propose Step-wise Representation adaPtation (SRP) and its variant SRP++, which start from a base forecasting model and add lightweight, step- or segment-specific encoder adaptations (via LoRA and a mixture-of-experts scheme) while keeping the linear head design unchanged. Experiments on several standard forecasting benchmarks and multiple backbone architectures report consistent improvements over the corresponding step-invariant baselines.

**Strengths:**

**1. Focused problem setup and theory.**
The paper isolates the issue of step-invariant representations in direct multi-step forecasting (not iterative/autoregressive), and gives a theorem showing an irreducible error for step-invariant representations with a linear decoder in the multi-step setting, which in presented analysis disappears in the single-step (t = 1) case; within this (admittedly narrow) setting the argument looks correct to me.

**2. Interesting representation vs step-structure idea.**
Explicitly questioning whether a single shared representation should serve all forecast steps is a worthwhile angle, and SRP/SRP++ are a clean way to introduce step-/segment-specific structure on top of existing models.

**3. Practical, modular method.**
SRP/SRP++ use LoRA-style adapters (and other PEFT variants) in a way that is easy to bolt onto iTransformer and other backbones. Training LoRA adapters per forecasting step (SRP), then moving to a soft MoE over adapters per segment (SRP++) is a natural iteration on the idea of step-/segment-specific adaptation. It’s nice that SRP++ is evaluated on multiple models beyond iTransformer, and that the ablation over adaptation techniques shows LoRA is a strong choice in this framework.

**4. Generally good presentation and empirical structure.**
The paper is clearly written, figures are mostly helpful, and the empirical section is well-structured around specific questions (performance, generality, flexibility, sensitivity). Apart from a few minor issues (figure/table placement, duplicated reference; details in Weaknesses), the presentation is good.

**Weaknesses:**

**1. Linear-decoder-only scope and missing discussion of neural decoders.**
All theory is built around a linear decoder acting on a single representation. There is no real discussion of neural/non-linear decoders (which are standard in text[1]/vision[2] and relevant for time-series[3]), even though such decoders could in principle create step-wise structure inside the decoder. This makes the setting feel quite limiting, and weakens the broad claim that the problem cannot be “overcome simply by advancing neural architectures.”

**2. Slippery boundary between representation and decoder.**
Conceptually, I find it more natural that the input time series has a single representation (it’s the same series regardless of the horizon), and that step-specific structure is created by the decoder. In SRP/SRP++, adapting the final part of the encoder per step/segment effectively “moves back” the step-invariant representation and makes the last encoder blocks act like a decoder. So the framing “step-invariant representations are the bottleneck” is somewhat problematic: you still have a step-invariant representation somewhere, and the adapters are effectively a more expressive decoder.

**3. Unclear connection between theorem and SRP++, and confounding with capacity.**
For SRP (pure per-step LoRA) the connection to step-specific representations is straightforward. For SRP++, however, representations are shared within segments via a soft MoE over shared experts (expert-weighted average per segment), and SRP++ also activates the most parameters per segment among the compared adaptation methods. This makes it unclear (to me) whether improvements are really due to “fixing the step-invariant bottleneck” or simply increasing model flexibility/parameter count; the original theoretical motivation feels less tightly linked at this point.

**4. Overstated empirical claims vs the actual numbers.**
Statements like “Overall, SRP++ significantly enhances the performance of iTransformer” and claims that SRP++ goes “beyond architectural designs” feel too strong given the reported gains. In particular, the highlighted case where iTransformer+SRP++ surpasses previous models on PEMS08 (MSE/MAE) is just a single instance; I don’t think it supports such broad conclusions.

**5. Weak “foundation model” story and missing FM literature.**
The paper mentions "foundation" models (abstract, sections 3.3 and 3.4), but there is no proper “foundation model” experiment: no multi-dataset pretraining with a separate evaluation suite, and no test on an existing pretrained time-series model (e.g., Chronos-style [3]). As is, I would either expect such experiments or a much more modest claim. There is also lack of citations / discussion of what is currently prevalent in the time-series foundation-model community.

**6. Minor presentation issues.**
Table 2 would be more readable if placed closer to the first paragraph of Section 4.3; I am not convinced Figure 3 adds much to motivation, performance comparison, or methodology (it might belong in the appendix); and there appears to be a duplicated Houlsby et al. (2019) reference in the bibliography. These are small issues but worth fixing.


[1] Sutskever, Ilya, Oriol Vinyals, and Quoc V. Le. "Sequence to sequence learning with neural networks." Advances in neural information processing systems 27 (2014).

[2] Ronneberger, Olaf, Philipp Fischer, and Thomas Brox. "U-net: Convolutional networks for biomedical image segmentation." International Conference on Medical image computing and computer-assisted intervention. Cham: Springer international publishing, 2015.

[3] Ansari, Abdul Fatir, et al. "Chronos: Learning the Language of Time Series." Transactions on Machine Learning Research.

**Questions:**

1. Since all theory is for a linear decoder, how do you see the argument extending (or not) to neural decoders? Do you think a sufficiently expressive, step-invariant neural decoder can still suffer from a similar irreducible-error phenomenon, or is your claim intentionally restricted to the linear-head case?
2. With segment-wise soft MoE, representations are shared within segments through mixtures of shared experts. In that setting, do you still consider the representation truly step-specific in the sense of your theorem, or only segment-specific? How do you reconcile the theorem’s step-invariant bottleneck with this partial sharing and the increased parameter count?
3. In the adaptation-method comparison, SRP++ uses soft MoE and because of that it appears to activate more parameters per segment than alternatives. Can you clarify to what extent the gains might just come from extra capacity? Is there any evidence (even qualitative) that similar-capacity baselines fail while SRP++ succeeds?
4. In the sensitivity study, relatively small k (few segments) already works well, which suggests that full per-step specialization is not actually required for good performance. How do you interpret this result relative to the claim that step-wise representations are critical? Does it support the view that the framework is effectively repurposing part of the encoder into a more expressive decoder acting on a still step-invariant input representation?
5. Given the current experiments, are you planning to either:
   -  add at least one “real” foundation-model-style experiment (e.g., multi-dataset pretraining and evaluation, or applying SRP/SRP++ to an existing pretrained time-series model), or
   -  tone down the foundation-model language and present this more modestly as a method for single-dataset/supervised direct forecasting?

I would be open to moving into the accept range if the mentioned issues are addressed convincingly.

---

### Official Review · Reviewer_aHfn · 2025-11-07

**Soundness:** 1
**Presentation:** 2
**Contribution:** 1
**Rating:** 2
**Confidence:** 3

**Summary:**

The paper argues that there exists a fundamental expressiveness bottleneck that limits the ability of step-invariant encoder-decoder type models of minimization of the forecasting error.  It introduces SRP and SRP++, methods which rely on 1) pre-training a model backbone in a one-step ahead fashion to avoid the bottleneck, then 2) introduce step-dependent LoRA heads and fine-tune them for long horizon forecasting. Authors empirically show that the introduced method slightly boosts the results, however, the extent of that boost is questionable.

Unfortunately, the paper, in my opinion, is very poorly written, and I found it hard to read due to insufficient explanations of introduced concepts, especially around expresiveness bottleneck. There are substantial issues with the claims made in the paper, as well as numerous minor yet relevant problems.

**Strengths:**

1. Introducing an interesting argument about expressiveness bottleneck, which is a theoretical limitation in form of a minimum forecasting error.
2. A set of interesting experiments
3. Making the code public  (though I haven't run it)

**Weaknesses:**

Minor:

1. Why introduce SRP and SRP++.  In my opinion it is confusing and SRP is a special case of SRP++.   A coherent story based around SRP++ would read better. Moreover,  SRP is not getting almost any coverage in the experiments except showing it in Tab. 2. I recommend re-evaluating that choice.

1. Lines 078,081: What is the reason for including these citations? They seem to be rather random and do not contribute anything. Wang et al. 2025 refers to a paper about forecasting in frequency domain while it is cited for predicting forecasts in a single pass.

1. Line 081-082: The notation here is confusing, why historical sequence is denoted with L while also the historical window length is L?

1. Line 090-092:  This statement is not true. There is plenty of research in decoder and encoder-decoder architectures. While I disagree with this claim, from a broader perspective, it is not relevant to state that in the paper since authors are not changing this claimed paradigm.

1. Lines ~122-147:  Figure 1 and the explanation seem insufficient. Why are these plots demonstrating that enforcing step-invariant representations create an expressiveness bottleneck? This is not explained.

1. Baselines: I think authors should also compare to non-transformer models that are usually state-of-the-art (e.g., NBEATS [3], NHITS [4]) and also, for reference, some statistical baselines (e.g., ARIMA, ETS).

1. ~Lines 203: It is never  clearly explained how exactly SRP overcomes the expressiveness bottleneck - it is simply stated as true. An explanation is necessary to give the reader some intuition.

Major:

1. 4.3. I appreciate reporting of 95% confidence intervals. However, authors argue incorrectly analyze the results (starting ~ line 417). CIs are very wide and largely overlapping, this doesn’t validate  the claim of SRP++  improving the results over baseline and especially not the claim of the number over 4%.  This is a clear statistical malpractice since the correct conclusion is that there is no significant difference between SI and SRP++.

    Moreover, since other adaptation techniques are reported in Table 2 I think they should also be  incorporated in Figure 4 for fair comparison. I think having that comparison would strongly benefit the analysis.


1. I believe that the existence of the expressiveness bottleneck is not sufficiently shown in the paper. We simply don’t know how big of an issue it is. Why is it not at least empirically quantified for different architectures and datasets? Moreover, in the main paper text there is little to no explanation on what the theorem is about and why it matters for the paper. This part of the paper is very poorly written.

   The entire expressiveness bottleneck argument (Theorem 3.1) is a trivial linear algebra claim that ignores the nature of representation learning. The theorem proves that a rank-L matrix cannot perfectly reconstruct an arbitrary, full-rank vector in a T-dimensional space (T>L). However, in my opinion this is highly likely not a relevant claim.  A forecast is not an arbitrary vector, but a highly structured, correlated signal with, probably, a low intrinsic dimensionality. The proof is not very applicable because it critiques the model's inability to generate full-rank noise, which is not the forecasting task.

1. 4.2 & Table 1: Results in iTransformer paper for the same experiment setup are slightly better for iTransformer than those reported here. While they may simply be due to randomness, it raises questions about the stability of the reported results in this experiment as well as the choice of hyperparameter tuning, especially since no other descriptive statistics are reported (e.g., std, se). I also crosschecked the results to the ones reported in Informer paper. It seems that the results reported by authors for Informer are worse by a considerable margin than what was reported in the Informer paper, but it is not clear if the experiment setup is exactly the same. I am concerned that the comparison and hyperparameter tuning is not fair.

    It would be useful to show that these gains are not simply because of the increase in parameter counts or simply from the optimization being smoother. I’m not convinced by the results that what matters is the step-dependent decoding.

    While the authors (in the appendix) claim a marginal parameter increase, they fail to prove that their performance gain isn't solely from this increase. They do not isolate the effect of the new architecture from the more favorable optimization landscape of fine-tuning. Ablations are necessary.

1. Line 043-45: The reason why contemporary approaches generate forecasts in one step is due to results from e.g., TimesNet [1] and Informer [2] that show error aggregation when autoregressively generating the forecast. However, authors do not relate their step-dependent representation to an autoregressive forecast generation which doesn’t have step-invariant representations.


[1] Wu et al. 2022 https://arxiv.org/abs/2210.02186.

[2] Zhou et al. 2020 https://arxiv.org/abs/2012.07436.

[3] Oreshkin et al. 2019 https://arxiv.org/abs/1905.10437.

[4] Challu et al. 2022 https://arxiv.org/abs/2201.12886.

**Questions:**

Please refer to the numerous questions raised in the Weaknesses section.

---

### Meta-Review · Area_Chair_xa5e · 2026-01-07

**Summary:**

The paper studies multi-step direct time-series forecasting with encoder/linear decoder architectures where a single, step-invariant representation is used to predict all future steps. All 5 reviewers are leaning towards rejection, with reasonable concerns that were not adequately addressed in the rebuttal phase. Therefore, I recommend rejection of this paper. I encourage the authors to revise the paper and resubmit it to future conferences.

**Reviewer Concerns:**

Reviewers mainly raised concerns on the minor improvements over baselines, narrow comparison scope with missing SOTA baselines, insufficient connection between theoretical analysis and model design, and presentation issues.

Authors can consider the suggestions from reviewers to improve the paper accordingly.

**Reviewer Scores:**

Given that the authors did not provide valid responses to the reviewers' concerns, I think the reviewers' scores will remain the same.

---

### Decision · Program_Chairs · 2026-01-26

Reject